# Comprehensive genomic analysis of dietary habits in UK Biobank identifies hundreds of genetic associations

Joanne B. Cole [1,2,3], Jose C. Florez[1,2,4] & Joel N. Hirschhorn[1,3,5] ✉

Unhealthful dietary habits are leading risk factors for life-altering diseases and mortality. Large-scale biobanks now enable genetic analysis of traits with modest heritability, such as diet. We perform a genomewide association on 85 single food intake and 85 principal component-derived dietary patterns from food frequency questionnaires in UK Biobank. We identify 814 associated loci, including olfactory receptor associations with fruit and tea intake; 136 associations are only identified using dietary patterns. Mendelian randomization suggests our top healthful dietary pattern driven by wholemeal vs. white bread consumption is causally influenced by factors correlated with education but is not strongly causal for coronary artery disease or type 2 diabetes. Overall, we demonstrate the value in complementary phenotyping approaches to complex dietary datasets, and the utility of genomic analysis to understand the relationships between diet and human health.

[1] Programs in Metabolism and Medical and Population Genetics, The Broad Institute of MIT and Harvard, Cambridge, MA, USA. [2] Diabetes Unit and Center for Genomic Medicine, Massachusetts General Hospital, Boston, MA, USA. [3] Division of Endocrinology and Center for Basic and Translational Obesity Research, Boston Children's Hospital, Boston, MA, USA. [4] Department of Medicine, Harvard Medical School, Boston, MA, USA. [5] Department of Genetics, Harvard Medical School, Boston, MA, USA. ✉email: Joel.Hirschhorn@childrens.harvard.edu

Unhealthful dietary habits are thought to be the leading risk factor for mortality both globally[1] and in the United States[2]. Overall, incidence rates of these dietary risk factors and their related diseases, like obesity and type 2 diabetes (T2D), are rising in parallel worldwide[3,4], causing global epidemics that require our urgent attention. Describing a biological basis for unhealthful dietary preferences could guide more effective dietary recommendations.

There is a clear, albeit modest, genetic component to diet, such as traditional measures of macronutrient intake (i.e., proportion of carbohydrate, fat, and protein to total energy intake), as demonstrated by significant heritability and individual genetic associations[5–9]. Five genome-wide association studies (GWAS) of macronutrient intake have been conducted to date; the most recent multi-trait analysis identified 96 independent genetic loci by combining summary statistics of individual macronutrient GWAS from 24-h diet recall questionnaires in 283K individuals in UK Biobank (UKB)[6–9]. In addition, the Neale Lab conducted GWAS (http://www.nealelab.is/uk-biobank/) across thousands of mostly binary traits analyzed primarily as dichotomous outcomes (i.e., wholemeal bread vs. all others) in 361K unrelated individuals in UKB. The recent GeneAtlas[10] improved power by using linear mixed models in 450K individuals in UKB, but analyzed a smaller set of dietary variables.

Additional measures of dietary intake, including both curated measures of single food intake (FI) and multivariate dietary patterns (DPs), such as those described by principal component (PC) analysis, have also shown significant associations with health outcomes in both epidemiological studies[11,12] and clinical trials[13]. Thus, with the recent advent of large biobank-sized population cohorts with dietary data, we can now perform GWAS with multiple complementary phenotyping approaches to examine a wide array of dietary habits, including previously unstudied single food comparisons (i.e., wholemeal vs. white bread) and DPs. Here, we report heritability and GWAS analysis (using linear mixed models) of both single FI, analyzed as curated single FI quantitative traits (FI-QTs), and of PC-derived DPs (PC-DPs) using Food Frequency Questionnaire (FFQ) data in up to 449,210 Europeans from UKB; we highlight association at biologically interesting loci (olfactory receptor loci associated with tea and fruit consumption) and use Mendelian randomization (MR) analyses to elucidate potentially causal relationships pertaining to specific dietary habits.

## Results

**Most dietary habits are correlated and heritable**. We derived 85 curated single FI-QTs from FFQ administered in UKB, using 35 nested and complementary questions (Supplementary Data 1; see Methods). As expected, given the nested nature of these questions, many pairs of the 85 FI-QTs were significantly correlated ($P < 0.05/85 = 5.88 \times 10^{-4}$; Supplementary Fig. 1 and Supplementary Data 2). We therefore also conducted PC analysis of these FI-QTs to generate 85 PC-DPs that capture correlation structure among intake of single foods and represent independent components of real-world dietary habits. The top 20 PC-DPs have eigenvalues >1, and explain 66.6% of the total variance in the 85 FI-QTs (Supplementary Fig. 2). While the inclusion of highly correlated FI-QTs in the PCA will affect the structure of the resulting PCs, it will not substantially affect the heritability (beyond additional noise when including multiple correlated variables) nor will it exclude FI-QTs that contribute significantly to the variance in the total FFQ dataset. Therefore, we used heritability of all 85 FI-QTs and 85 PC-DPs as a filter for downstream genomic analysis.

Overall, 84.1% of dietary habits analyzed (83/85 FI-QTs and 60/85 PC-DPs) were significantly heritable (as assessed by $h_g^2$ $P < 0.05/170 = 2.9 \times 10^{-4}$; see Methods, Table 1, Supplementary Data 1, and Supplementary Fig. 3), displayed extensive genetic correlation ($r_g$; Supplementary Data 3), and were included in downstream genomic analysis. Phenotypic and genetic correlation between the 60 PC-DPs and their significantly contributing FI-QTs are illustrated in Supplementary Fig. 4. The most heritable FI-QTs fall into a handful of dietary food groups related to milk consumption, alcohol intake, and butter/spread consumption. The first PC-DP (hereafter referred to as PC1) is among the most heritable DPs (PC1 $h_g^2 = 13.6\%$, Table 1), and is more heritable than all of its individual contributing FI-QTs (all $h_g^2$ comparisons have $P < 5.0 \times 10^{-45}$, see Methods), including bread type (max $h_g^2 = 9.5\%$, Supplementary Data 1).

**Table 1 SNP heritability estimates $\left(h_g^2\right)$ and number of significant GWAS loci for the most heritable 20 dietary habits.**

| Dietary habit | N | $h_g^2$ | SE | P value | Number of significant loci |
|---|---|---|---|---|---|
| Milk type: soy milk vs. never | 31,889 | 0.282 | 0.010 | $5.8 \times 10^{-175}$ | 1 |
| Milk type: full cream vs. never | 43,995 | 0.243 | 0.007 | $4.8 \times 10^{-264}$ | 1 |
| Spread type: tub margarine vs. never | 77,738 | 0.182 | 0.004 | $<1.00 \times 10^{-300}$ | 5 |
| Among current drinkers, drinks usually with meals: yes vs. no | 235,312 | 0.160 | 0.001 | $<1.00 \times 10^{-300}$ | 30 |
| PC1 | 449,210 | 0.136 | 0.001 | $<1.00 \times 10^{-300}$ | 140 |
| Spread type: flora + benecol vs. never | 86,823 | 0.133 | 0.004 | $2 \times 10^{-242}$ | 2 |
| Milk type: other milk vs. never | 20,557 | 0.128 | 0.015 | $1.42 \times 10^{-17}$ | 0 |
| Overall alcohol intake | 448,623 | 0.121 | 0.001 | $<1.00 \times 10^{-300}$ | 98 |
| PC3 | 449,210 | 0.118 | 0.001 | $<1.00 \times 10^{-300}$ | 82 |
| Glasses of water per day | 445,965 | 0.116 | 0.001 | $<1.00 \times 10^{-300}$ | 78 |
| Total drinks of alcohol per month | 449,210 | 0.113 | 0.001 | $<1.00 \times 10^{-300}$ | 104 |
| Spread type: low fat spread vs. never | 72,017 | 0.110 | 0.004 | $1.8 \times 10^{-166}$ | 0 |
| Coffee type: decaffeinated vs. any other | 353,710 | 0.108 | 0.001 | $<1.00 \times 10^{-300}$ | 36 |
| Pieces of fresh fruit per day | 447,401 | 0.108 | 0.001 | $<1.00 \times 10^{-300}$ | 99 |
| Overall cheese intake | 438,453 | 0.108 | 0.001 | $<1.00 \times 10^{-300}$ | 60 |
| Spread type: butter vs. never | 213,549 | 0.102 | 0.001 | $<1.00 \times 10^{-300}$ | 15 |
| Frequency of adding salt to food | 448,890 | 0.102 | 0.001 | $<1.00 \times 10^{-300}$ | 82 |
| Among current drinkers, drinks usually with meals: yes/, it varies, no | 357,136 | 0.101 | 0.001 | $<1.00 \times 10^{-300}$ | 29 |
| Bread type: white vs. wholemeal/wholegrain + brown | 416,312 | 0.100 | 0.001 | $<1.00 \times 10^{-300}$ | 46 |
| Milk type: skimmed vs. never | 108,035 | 0.098 | 0.003 | $<1.00 \times 10^{-300}$ | 1 |

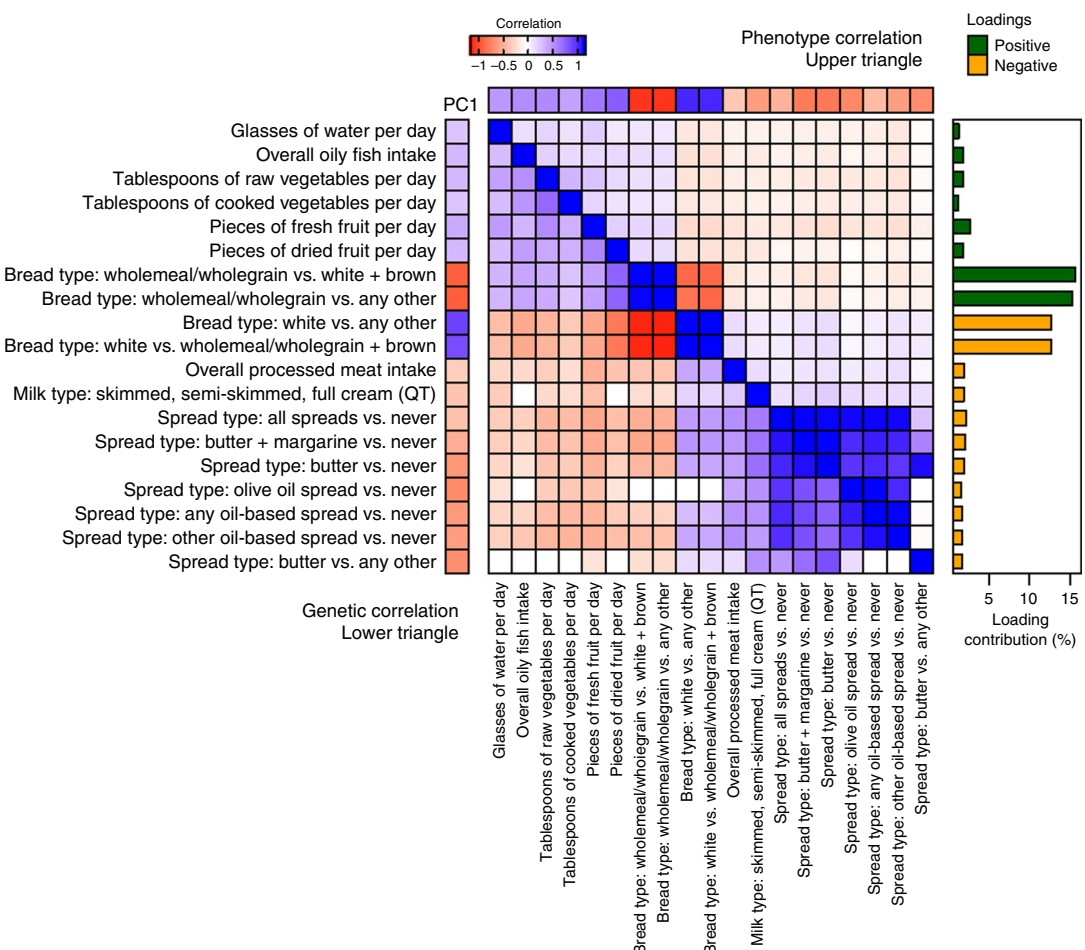

**Fig. 1 Relationships between PC1 and its 19 significantly contributing single food intake QT.** The heat map depicts the phenotypic (upper triangle) and genetic (lower triangle) correlation between the 19 significantly contributing single FI-QTs with each other and PC1. All correlations with nonsignificant *P* values (*P* > 0.05/85) were set to 0. Percent contribution of each of the 19 traits to PC1 is depicted in the correlation matrix bar plot annotation on the right, colored by loading direction.

PC1, which explains 8.63% (Supplementary Fig. 2) of the total phenotypic variance in FI-QTs, contains foods similar to those that make up previously described Western and prudent dietary factors[14], although our PC1 is primarily defined by the type of bread consumed (wholegrain/wholemeal vs. white bread). Overall, the FI-QTs that have significant positive loadings for PC1 include wholemeal/wholegrain bread consumption (two correlated FI-QTs contributing 15.4–15.8%), increased fruit and vegetable intake (four correlated FI-QTs contributing 1.2–2.6%), increased oily fish intake (1.8%), and increased water intake (1.2%). The FI-QTs that have significant negative loadings include white bread consumption (two correlated FI-QTs contributing 12.7–12.8%), butter and oil spread consumption (seven correlated FI-QTs contributing 1.5–2.2%), increased processed meat intake (1.9%), and consumption of milk with higher fat content (1.8%; Fig. 1).

**Dietary habit GWAS in UKB identifies 814 independent loci.** GWAS on the 143 significantly heritable dietary habits, using linear mixed models in up to 449,210 individuals, identified 814 independent loci (defined as >500 kb apart) surpassing genome-wide significance ($P < 5.0 \times 10^{-8}$). Of these, 309 also surpass a more conservative Bonferroni-corrected study-wide significance threshold ($P < 5.0 \times 10^{-8}$/143 traits = $2.9 \times 10^{-10}$; Supplementary Data 4). Across the 143 dietary traits, there was a clear positive correlation of heritability estimates ($h_g^2$) with the number of

significant loci and the variance explained by these loci (Fig. 2), a pattern that persisted when limited to phenotypically independent DP-PCs, with outliers often explained by smaller sample size (Supplementary Fig. 5). We also found evidence that the more heritable phenotypes have led to single-nucleotide polymorphisms (SNPs) with larger effect sizes rather than simply a greater number of significant loci (Fig. 2 and Supplementary Fig. 5). Notably, there is a mix of FI-QTs and PC-DPs among the most successful GWAS traits, with the PC1 GWAS identifying the most significant loci ($M = 140$) that together explain the largest amount of variance of any dietary habit analyzed (6.65%).

Recent work in a non-mixed model setting has demonstrated the presence of latent structure in UKB co-incident with outcomes such as educational attainment, which is also associated with dietary intake[15,16]. We found that our linear mixed model dietary habit GWAS had little to no residual confounding as estimated by linkage disequilibrium (LD) score regression (LDSC) intercepts (median = 1.018, inter-quartile range (IQR): 1.011–1.030) and ratios (median = 0.087, IQR: 0.059–0.0123; Supplementary Data 1). Furthermore, large-scale sensitivity analysis re-conducting all 143 significantly heritable dietary habit GWAS additionally adjusting for assessment center in the genetic model only found slight attenuation of heritability (median from 4.6% to 4.2%, IQR from 2.4–7.9% to 2.2–7.4%). We report the results in this paper without additional adjustment for assessment center.

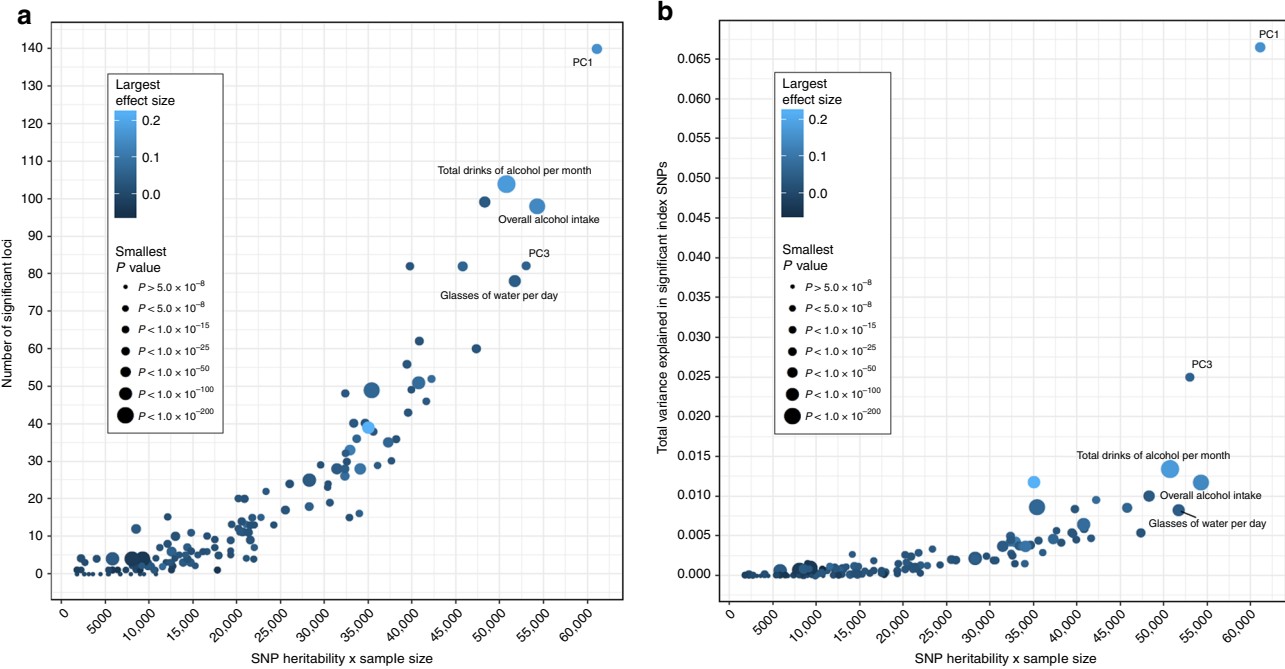

**Fig. 2 Relationship between SNP heritability and GWAS success. a** Scatter plots of the number of genome-wide significant loci or **b** variance explained by genome-wide significant SNPs vs. SNP heritability × sample size for all 143 significantly heritable traits. Points are colored by the largest effect size and sized by the smallest $P$ value of their significant index SNPs.

Of the 814 index SNPs, 767 (94%) are common with minor allele frequencies ≥5%; similar to previous GWAS findings[17], the vast majority are either intronic or intergenic (82.2%). Credible set analysis found that more than one-third of our lead signals (289/814) were driven by 10 or fewer SNPs, 124 had 2–5 credible set SNPs and 54 had a single 95% credible set SNP (see Methods). Of the credible sets with single SNPs, 65.2% are intronic or intergenic, while 13.0% are missense variants and the remainder largely regulatory (by comparison, only 1.9% of index SNPs from multi-SNP credible sets are missense). We investigated whether any of our loci were previously reported at genome-wide significance for any trait in either the GWAS catalog[18] or the comprehensive Neale Lab GWAS (http://www.nealelab.is/uk-biobank/) of 4358 traits in 361,194 unrelated individuals in UKB. Of the 814 dietary habit GWAS loci, 205 have never been previously reported. This can largely be explained by our use of previously unstudied curated FI-QTs and PC-DPs and/or statistical power gained from linear mixed models in nearly 450K individuals.

**Olfactory receptor loci are associated with specific foods.** Among the lead loci are several regions containing clusters of olfactory receptor genes. Although our dietary habit GWAS loci are not enriched for olfactory receptor genes (Fisher's test $P = 0.419$), the association of olfactory receptors with specific FI-QTs supports the well-known link between smell and taste. The top signal associated with "pieces of fresh fruit eaten per day" is a region on chromosome 7q35 (lead SNP rs10249294 $P = 5.7 \times 10^{-65}$; Supplementary Fig. 6). All 44 SNPs within the 95% credible set cover a 27 kb region containing olfactory receptor gene *OR6B1* (Supplementary Fig. 7). Another locus specific to fruit intake, on chromosome 14q (Supplementary Fig. 6), is associated with both "pieces of fresh fruit per day" (rs34162196 $P = 5.1 \times 10^{-24}$) and "pieces of dried fruit per day" (rs35260863 $P = 6.4 \times 10^{-18}$). The only two SNPs contained in both 95% credible sets cover a 5.8 kb region (chr14:22,038,125–22,043,949)

containing olfactory receptor gene *OR10G3* (Supplementary Fig. 7).

Five additional loci have 95% credible sets that overlap olfactory receptor genes and are strongly associated with a single FFQ question. Notably, "cups of tea per day" is associated with a region on chr11q12 (lead SNP rs1453548 $P = 3.1 \times 10^{-9}$) that contains six different olfactory receptor genes, including *OR5AN1*, *OR5A2*, *OR5A1*, *OR4D6*, *OR4D10*, and *OR4D11* (Supplementary Fig. 7). The 95% credible set contains SNP rs7943953 (chr11:59,224,144) previously reported for being associated with odor perception, and specifically that of β-ionone (floral) sensitivity, an aroma compound found in high quantities in tea[19–21]. By contrast, "cups of coffee per day" is associated with a nearby yet distinct region on chr11q12 (lead SNP rs643017 $P = 1.1 \times 10^{-8}$) in a dense cluster of olfactory receptor genes containing *OR8U8* and *OR5M3* (Supplementary Fig. 7). Other associations with olfactory receptor gene regions include regions containing *OR52J3* and *OR52E2* with three dietary habits measuring butter consumption (rs2445249 with "spread type: butter vs. any other" $P = 9.1 \times 10^{-15}$), *OR4K17* with "tablespoons of raw vegetables per day" (rs9323534, $P = 4.1 \times 10^{-10}$), and *OR10A3* and *OR10A6* with "overall cheese intake" (rs757969034, $P = 2.4 \times 10^{-8}$; Supplementary Fig. 7).

**GWAS of dietary pattern PC1.** Of the PC-DPs, the PC1 DP has the highest heritability and most genome-wide significant loci, and previously identified DPs with similar food make-ups (although with different factor loadings) have been associated with disease[22,23]. Although PC1 is both phenotypically and genetically correlated with its 19 contributing FI-QTs (absolute $r_p = 0.25–0.81$; absolute $r_g = 0.29–0.93$), the GWAS results provide distinct sets of significant associations. Together, PC1 and its 19 contributing traits are associated with a total of 387 independent genome-wide significant loci, falling into one of four categories (Fig. 3): 55 loci significant for PC1 only (dark blue), 282 loci significant for one or more of the 19 contributing FI-QTs only (dark red), 37 loci more significant for PC1 (light blue), and

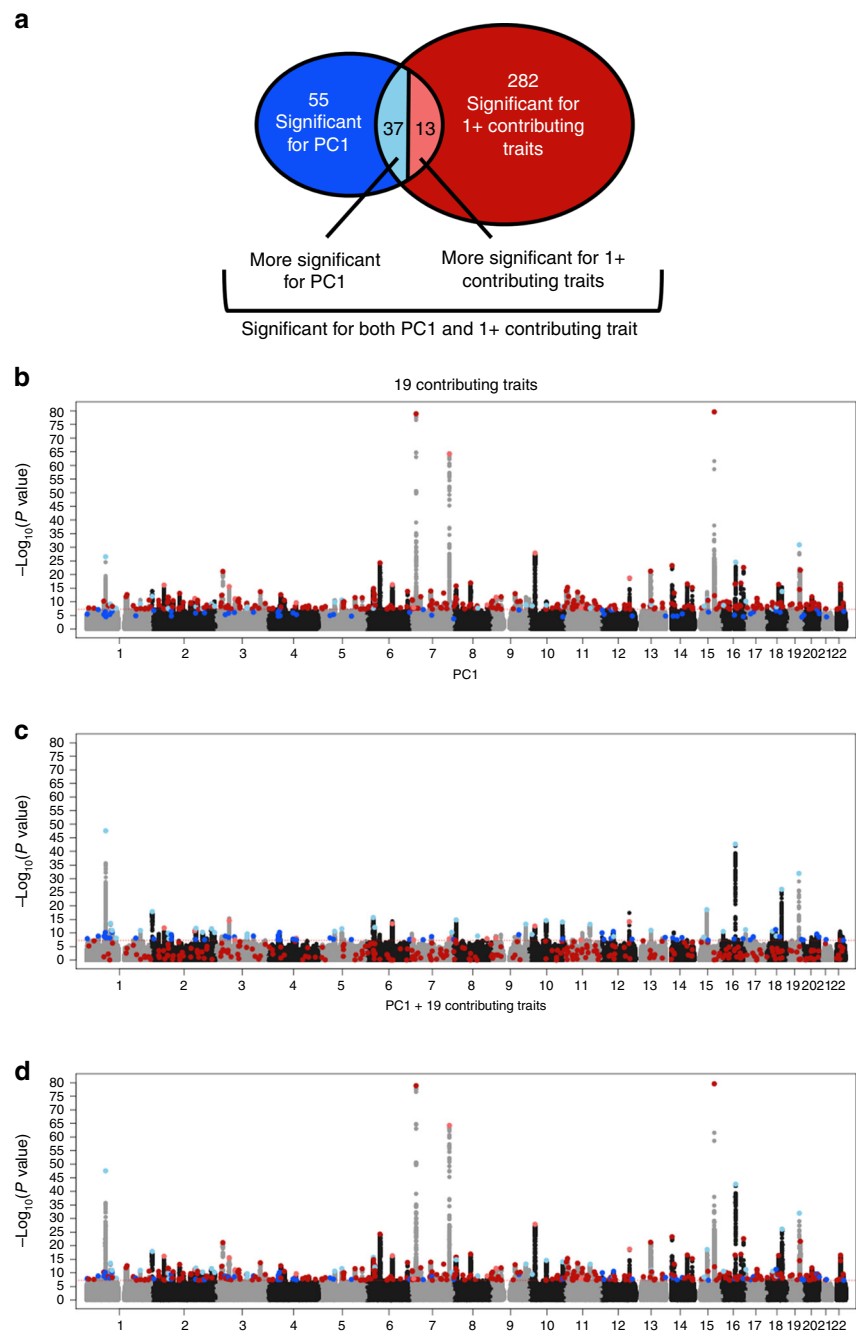

**Fig. 3 Manhattan plot of PC1 and its 19 significantly contributing single food intake QT. a** Schematic depiction of four groups of significant loci (and corresponding **b, c** Manhattan plot colors) for various combinations of PC1 and its 19 contributing traits. Fifty-five loci significant for PC1 only (dark blue), 282 loci significant for one or more of the 19 contributing FI-QTs only (dark red), 37 loci more significant for PC1 (light blue), and 13 loci more significant for one or more of the 19 FI-QTs (light red). **b** Combined Manhattan plot of the minimum P value across the 19 FI-QTs that significantly contribute to PC1. **c** Manhattan plot of PC1 only. **d** Combined Manhattan plot of the minimum P value across PC1 and the 19 FI-QTs.

13 loci more significant for one or more of the 19 QTs (light red). The 55 loci significantly associated with PC1 but not FI-QTs still trend toward association with one or more of the 19 contributing FI-QTs, whereas the reciprocal is not always the case: some FI-QT-associated SNPs display no association with PC1 at all. This observation indicates that the use of PCs can increase power to detect some associations, while others are only detectable through association with specific foods, supporting the use of both of these complementary phenotyping approaches to more effectively define the genetic architecture of dietary intake.

In addition to being correlated with its contributing FI-QTs, PC1 displays significant genetic correlation with 248 non-diet-related traits, including traits relating to physical activity, educational attainment, socioeconomic status, smoking status, medication codes, and urine biomarkers (Supplementary Data 5). The lead PC1 SNP, rs66495454, is a common indel (minor allele frequency (MAF) = 38%) located at chr1:72,748,567 in the promoter of neuronal growth factor 1 (*NEGR1;* Supplementary Fig. 7). SNP rs66495454's deletion allele (−/TCCT) is associated with a decrease in PC1, indicating a shift towards white bread

consumption and other unhealthful foods ($\beta = -0.017$, $P = 2.80 \times 10^{-48}$) and has been previously reported as associated with a decrease in intelligence, educational attainment, and, perhaps surprisingly, body mass index (BMI)[24–26]. We were struck by the observation that 122 PC1-associated loci were already present in the GWAS catalog[18] or Neale Lab GWAS (http://www.nealelab.is/uk-biobank/), and of these, 83 were associated with intelligence or cognitive ability, educational attainment, or obesity-related anthropometric traits, including rs1421085 in *FTO* and rs429358 in *APOE* (Supplementary Data 4). Of these 83, 24 were associated with intelligence or educational attainment and not obesity-related traits; 38 were associated with obesity-related traits and not intelligence or educational attainment, indicating PC1 may be responsive to distinct sets of heritable factors related separately to educational attainment and to BMI. DEPICT (Data-driven Expression Prioritized Integration for Complex Traits) pathway analysis of PC1 identified enrichment of two gene sets, including axonogenesis, and 22 tissues, of which 21 were brain related (false discovery rate <0.05; Supplementary Data 6). The large and broad overlap in both the significant loci and the overall genetic make-up of our dietary habits with other traits, many of which do not have well-established biological links with diet, emphasizes a need for exploring correlation vs. causation.

Therefore, we sought to understand the cause-and-effect relationships and degree of pleiotropy between PC1 and educational attainment, fluid intelligence scores, and BMI using bidirectional MR[27]. We performed additional GWAS of educational attainment, fluid intelligence scores, and BMI in UKB and applied bidirectional MR with these phenotypes and PC1 (see Methods). Both inverse variance weighted (IVW) random-effects and weighted median (WM) MR provided significant evidence of causal effects of educational attainment and fluid intelligence on PC1 (educational attainment WM $\beta = 0.823$, 95% confidence intervals [CI]: 0.751–0.894, $P = 6.59 \times 10^{-113}$; intelligence WM $\beta = 0.261$, 95% CI: 0.190–0.332, $P = 7.71 \times 10^{-13}$; Table 2 and Supplementary Fig. 8). Egger regression intercepts for both intelligence and educational attainment were not significantly different than 0 (Table 2) and causal effect estimates were essentially unchanged after different approaches to variant filtering (see Methods and Supplementary Table 1). For each 1 standard deviation increase in educational attainment and intelligence, these data estimate that PC1 shifts from white bread consumption and foods often described as unhealthful to wholemeal bread consumption and foods often described as healthful by 0.823 and 0.261 standard deviations, respectively. To rule out weak instrument bias in overlapping samples, we validated our educational attainment results in two ways with high congruence. An externally derived educational attainment genetic instrument[26] had an effect estimate on PC1 equal to 0.708 (95% CI: 0.576–0.840, $P = 6.23 \times 10^{-26}$), and splitting UKB into training and testing datasets also yielded similar and significant effect estimates (Supplementary Table 2). Results were also in high agreement with a repeated MR analysis using our sensitivity analysis PC1 GWAS results adjusted for assessment center (Supplementary Table 2). While PC1 is largely driven by bread type, MR of educational attainment on PC1's 19 contributing FI-QTs demonstrates significant causal effects on both bread and non-bread FI-QTs, suggesting that while smaller, the other foods contributing to PC1 have a significant impact (Supplementary Table 3).

The reverse analysis shows significant but smaller estimates of causal influences of PC1 on educational attainment and fluid intelligence (educational attainment WM $\beta = 0.199$, 95% CI: 0.175–0.224, $P = 7.44 \times 10^{-56}$; intelligence WM $\beta = 0.074$, 95% CI: 0.045–0.104, $P = 8.04 \times 10^{-7}$). The smaller effects suggest that while there could be pleiotropy undetected by the

**Table 2 Bidirectional Mendelian randomization results for PC1 as both an exposure and outcome with educational attainment (ED), fluid intelligence scores (INT), and BMI.**

| Exposure | Outcome | SNPs | Inverse variance weighted | | | | MR Egger | | | | | | Weighted median | | | |
|---|---|---|---|---|---|---|---|---|---|---|---|---|---|---|---|---|
| | | | Est. | CI Lower | CI Upper | P value | Est. | CI Lower | CI Upper | P value | Intercept | Int. P value | Est. | CI Lower | CI Upper | P value |
| ED | PC1 | 309 | 0.836 | 0.765 | 0.906 | $1.69 \times 10^{-118}$ | 0.970 | 0.681 | 1.259 | $4.88 \times 10^{-11}$ | −0.002 | 0.347 | 0.823 | 0.751 | 0.894 | $6.59 \times 10^{-113}$ |
| PC1 | ED | 140 | 0.203 | 0.173 | 0.232 | $2.20 \times 10^{-41}$ | 0.182 | 0.068 | 0.296 | $1.70 \times 10^{-3}$ | 0.001 | 0.715 | 0.199 | 0.175 | 0.224 | $7.44 \times 10^{-56}$ |
| INT | PC1 | 184 | 0.264 | 0.183 | 0.345 | $1.82 \times 10^{-10}$ | 0.080 | −0.260 | 0.420 | 0.644 | 0.004 | 0.276 | 0.261 | 0.190 | 0.332 | $7.71 \times 10^{-13}$ |
| PC1 | INT | 140 | 0.105 | 0.071 | 0.138 | $8.23 \times 10^{-10}$ | 0.100 | −0.029 | 0.229 | 0.129 | 0.000 | 0.941 | 0.074 | 0.045 | 0.104 | $8.04 \times 10^{-7}$ |
| BMI | PC1 | 1165 | 0.000 | −0.034 | 0.035 | 0.993 | 0.319 | 0.223 | 0.415 | $6.97 \times 10^{-11}$ | −0.007 | $3.30 \times 10^{-12}$ | 0.014 | −0.019 | 0.047 | 0.403 |
| PC1 | BMI | 140 | 0.089 | 0.012 | 0.166 | $2.34 \times 10^{-2}$ | 0.432 | 0.140 | 0.724 | $3.78 \times 10^{-3}$ | −0.012 | 0.017 | 0.007 | −0.019 | 0.033 | 0.614 |

Egger and WM approaches, or true bidirectional effects, the causal influences are more likely to be in the direction of educational attainment to wholemeal bread consumption and other healthful foods. Importantly, because the instrumental variables used for educational attainment are not mechanistically linked directly to educational attainment/intelligence, it remains possible that causal influences on PC1 could be due to unmeasured heritable factor(s) that are themselves causal for educational attainment/intelligence. Furthermore, because PC1 is highly correlated with hundreds of additional outcomes, educational attainment (or traits correlated with educational attainment) could be influencing a larger complex phenotype made of many lifestyle factors that is captured here by PC1 dietary preferences. In contrast to the educational attainment analyses, we were unable to provide robust evidence of a causal relationship in either direction between BMI and PC1 due to significant heterogeneous pleiotropic effects leading to inconsistent causal effect estimates (Table 2 and Supplementary Fig. 8). Taken together with an overlap between genome-wide significant variants and small overall genetic correlation ($r_g = -0.056$, $P = 0.024$), we postulate that BMI and our non-isocaloric PC1 have a highly complex multi-pathway relationship for which there may be overlapping genetic etiology, but a mostly distinct underlying genetic architecture.

**Causal relationships with disease and related risk factors.** To test whether the PC1 DP, driven largely by bread type, is capturing a lifestyle that is likely to causally influence disease risk, we repeated bidirectional MR analysis between PC1 with coronary artery disease (CAD) from the mostly European CARDIo-GRAMplusC4D GWAS and T2D from the DIAGRAM consortium 2017 GWAS[28,29]. The only association that provides robust evidence of a causal relationship was an increased risk in CAD leading to an increase in PC1, indicating wholemeal over white bread consumption, with a significant, albeit small effect (WM $\beta = 0.0458$, 95% CI: 0.016–0.076, $P = 0.003$, Supplementary Table 4), suggesting reverse causation of CAD on diet. Although we found that higher educational attainment increases PC1 wholemeal vs. white bread consumption and other healthful foods, using our genetic instruments we did not identify causal evidence that PC1, driven by bread type, causes a decreased risk for CAD or T2D. Again, similar results were found when using our sensitivity analysis PC1 GWAS results adjusted for assessment center (Supplementary Table 4).

In contrast to the associations with PC1, a single SNP, rs1453548, strongly influences "cups of tea per day" and has a plausible biological mechanism (it is located in an olfactory receptor-dense region and explains >96% of the observed phenotypic variance of β-ionone sensitivity[30]). We therefore performed an MR analysis using rs1453548 on the complete set of traits in UKB using the Neale Lab GWAS (http://www.nealelab.is/uk-biobank/). Using a strict Bonferroni-corrected significance threshold ($P < 0.05/4358$ traits $= 1.15 \times 10^{-5}$), we identified a significant causal effect of "cups of tea per day" on smoking status, for which increases in the minor allele T (MAF = 34%) that cause an increase in "cups of tea per day" cause a decrease in smoking status (Neale 20160:ever smoked Wald ratio estimate = −0.51, 95% CI: −0.69 to −0.32, $P = 4.326 \times 10^{-8}$; Supplementary Data 7). However, rs1453548 is directly associated with "ever smoked" smoking status in the Neale Lab GWAS at genome-wide significance ($P = 4.329 \times 10^{-8}$), the 51-SNP "cups of tea per day" genetic instrument excluding rs1453548 has no significant causal effect on smoking status (IVW $P = 0.20$), and β-ionone is also found in tobacco[31], suggesting the effects of rs1453548 on odor perception of β-ionone may have pleiotropic effects on both smoking status and tea drinking. Together with a lack of additional significant causal relationships between rs1453548 and health outcomes in UKB, our results indicate that drinking more tea does not have clear effects on health outcomes in UKB, and it is possible that some previous reports on the health benefits of drinking more tea are a result of confounding with smoking status.

**Discussion**

Understanding the genetic architecture of dietary habits has immense implications for human health, but has been a difficult task, in part due to the low heritability of many dietary traits. The recent advent of large-scale datasets such as UKB, with deep phenotyping on hundreds of thousands of individuals, has now made genetic discovery of traits with relatively low heritability possible. Expansion of phenotyping to include both curated FI-QTs and PC-DPs together with GWAS in nearly 450K individuals allowed our study to make hundreds of new genetic discoveries relating to diet. Our work advances the elucidation of the genetic architecture of multiple correlated dietary habits and helps lay the groundwork for future research on nutrigenomic and other complex multifactorial multivariable datasets.

Our work emphasizes the importance of interrogating the genetics of complementary phenotypes to glean a more complete picture of the genetic architecture of diet. One of the strongest associations we observed was between SNP rs1229984 and the FI-QT "total drinks of alcohol per month" ($P = 3.8 \times 10^{-248}$; Supplementary Fig. 6). This SNP has been previously associated with alcohol consumption[10,32–35]; consistent with a recent meta-analysis[35], our use of a curated and quantitative FI-QT improved power compared with the more categorical "overall alcohol intake" phenotype and individual alcohol subtypes (i.e., "red wine glasses per month"; Supplementary Figs. 6 and 9). This increase in power is consistent with the high genetic correlation but low phenotypic correlation between individual questions related to alcohol, indicating that a composite alcohol question is more suitable for genetic discovery (Supplementary Fig. 10). Furthermore, using PC1 as an example, we demonstrate that the genetic architecture of FI-QTs and PC-DPs are distinct, with hundreds of genetic associations more strongly associated with either PC1 or with its contributing FI-QTs. Overall, by using complementary phenotyping approaches, we identified 814 independent genetic associations, of which 205 were completely novel, 311 were uniquely associated with curated FI-QTs, and 136 were uniquely associated with PC-DPs.

Of note, PC-DPs described throughout our manuscript were derived from a data-driven and unbiased PCA that included all FI-QTs, including those that were highly correlated and based upon the same FFQ question. Although this approach will shift each PC towards any of its overrepresented correlated measures, as is seen with bread type, which makes up 56.7% of PC1, it will not substantially inflate heritability (other than by diminishing noise introduced by repeated measures of correlated phenotypes) nor will it exclude questions that significantly contribute to the variance in the total FFQ dataset. This also suggests that some PC-DPs will simply capture correlation between FI-QTs from the same FFQ question. For example, PC2 is made up of 11 significant FI-QTs based on a single FFQ question relating to the use of butter- and oil-based spreads (Supplementary Fig. 4). On the other hand, some PC-DPs capture the correlation between a variety of FI-QTs and FFQ questions fairly equally, such as PC7 whose lead contributors –3 to 7% each) include fish and meat intake, alcohol intake, tea and coffee consumption, cereal intake, fruit and vegetable intake, cheese intake, and never eating sugar (Supplementary Fig. 4). Not surprisingly, SNP heritability, the

proportion of phenotypic variance explained by common genetic variation, and number of GWAS loci is larger for PC7, a much more diverse DP, than for PC2 (Supplementary Fig. 3B). Notably, future work with DPs derived from non-overlapping FFQ questions could uncover new biology not discovered in our data-driven approach.

Of note, the interpretation of the genetic component of predominantly environmental traits, such as dietary intake, can be complicated. As we have shown, dietary habits are highly correlated both with each other and with non-dietary traits, suggesting that any single dietary phenotype may represent a broader diet and lifestyle. For instance, DP PC1 with both high genetic correlation with educational attainment and large overlap in genome-wide significant loci with BMI, is capturing variance in both obesity-related traits and measures of socioeconomic status. Additionally, similar to most nutritional epidemiology studies, measures of dietary intake in UKB are based on self-reported questionnaire data, which intrinsically suffers biases based on memory and favorable reporting[36,37], further complicating the interpretation of genomic results. As an initial exploration of the implications of genetically influenced composite DPs, we focused on the strong genetic overlap between PC1 DP and phenotypes related to educational attainment[38]. While bidirectional MR demonstrates some pleiotropic effects between educational attainment and PC1, indicating either shared biology or upstream common cause(s), the relative strengths of these causal estimates suggests that higher educational attainment and/or correlated phenotypes (such as socioeconomic status [income $r_g = 0.77$] or factors related to school performance, such as fluid intelligence test scores [$r_g = 0.68$]) either directly or indirectly shift towards wholemeal bread vs. white bread consumption, and to a much lesser extent other foods described as healthful. While previous observational studies have shown that Western and prudent DPs are associated with CAD and T2D[11,23], our MR analysis of PC1, driven by bread type, on CAD or T2D did not demonstrate a causal effect from diet to disease, but rather a small suggestion of a reverse causal relationship between CAD and diet (CAD diagnosis leads to more wholemeal vs. white bread consumption and higher intake of other healthfully described foods).

The conclusion that our PC1 DP does not appear to be a causal risk factor for disease must be viewed in the context of several potential limitations of our study. Genetic instruments derived from genome-wide significant variants tend to explain a small fraction of phenotypic variance, which can lead to lack of power to detect potentially true causal effects of diet on outcomes, although this is mitigated by the large sample size of the UKB cohort. Additionally, while the use of overlapping samples in MR could in theory lead to inflated causal estimates[39], UKB's large sample size provides robust genetic instruments, and the strength of our causal associations, combined with our validation analysis with an independently ascertained set of instruments for educational attainment, suggesting that our results are likely not influenced by weak instrument bias. Furthermore, although we did not detect evidence of pleiotropy, it remains possible that pleiotropic effects of some of the variants associated with PC1 masked a causal effect on cardiometabolic disease risk. Finally, PC1 may not capture the causal protective features of a prudent DP as it is largely driven by bread type. However, it remains possible that there is a stronger correlative than causal relationship between PC1 or the previously described Western DP and increased risk of cardiometabolic disease.

We also find several interesting associations between specific FI-QTs (fruit, tea, coffee, vegetables, cheese, and butter) and olfactory receptors. The chr11p15 locus controlling odor perception of β-ionone[30], described as smelling of cedar wood but upon dilution (e.g., in tea) a more floral aroma[40], has pleiotropic effects

that both reduce the chances of ever smoking and increase tea intake. While SNPs at chr11p15 have already been shown to be associated with food choice with and without added β-ionone[30], we highlight here for the first time a link between β-ionone odor perception with smoking status, with potential significant implications for smoking-related health problems. This result also highlights the importance of understanding the pleiotropic consequences of variants used as genetic instruments in MR.

Of note, our dietary habits derived from a shortened FFQ were not adjusted for total energy intake, a measure highly correlated with physical activity and body weight[41]; as such, our dietary habits represent potentially non-isocaloric variations in dietary intake. However, we found minimal phenotypic correlation between any of our FFQ-derived phenotypes and 24-h recall questionnaire-derived total energy intake (maximum correlation $r = 0.037$). Furthermore, none of our lead 814 SNPs were nominally significant in the Neale Lab total energy intake GWAS ($P > 0.05/814$). Nine of our phenotypes, including "slices of bread per week," "overall cheese intake," and "glasses of water per day" did show significant genetic correlations with total energy intake, suggesting that the genetic architecture of these traits could be shared with traits that reflect more global lifestyle and DPs (Supplementary Data 5).

Overall, we present the genetic analysis of two complementary phenotypic approaches for dietary habits in a well-powered sample. Our results expand the understanding of the genetic contributors to dietary preferences and highlight the advantages of using complementary and novel approaches to derive carefully curated phenotypes, both for diet and for polygenic traits more generally. Our results also empower investigations of how our eating habits causally relate to disease risk. Comprehensive and rigorous investigation into the causal consequences of different modifiable aspects of diet and lifestyle can potentially have enormous implications for public health.

## Methods

**UKB genetic data.** UKB is a large prospective cohort with both deep phenotyping and molecular data, including genome-wide genotyping, on over 500,000 individuals aged 40–69 years living throughout the UK between 2006 and 2010[42]. Genotyping, imputation, and initial quality control on the genetic dataset has been described previously[43,44]. Additionally, we removed individuals flagged for failing UKBiLEVE genotype quality control, heterozygosity or missingness outliers, individuals with putative sex chromosome aneuploidy, individuals with self-reported vs. genetically inferred sex mismatches, and individuals whom withdrew consent at the time of analysis. Work within was conducted on genetic data release version 3, with imputation to both Haplotype Reference Consortium and 1000 Genomes Project (1KGP)[45], under UKB application 11898. We have complied with all relevant ethical regulations for work with UKB and all participants provided informed consent.

**UKB phenotype derivation.** All phenotype derivation and genomic analysis was conducted on a homogenous population of individuals of European (EUR) ancestry ($N = 455,146$), as determined by: (1) projection on to 1KGP phase 3 PCA space, (2) outlier detection to identify the largest cluster of individuals using Aberrant R package[46], selecting the $\lambda$ in which all clustered individuals fell within 1KGP EUR PC1 and PC2 limits ($\lambda = 4.5$), (3) removed individuals who did not self-report as "British," "Irish," "Any other white background," "White," "Do not know," or "Prefer not to answer," as self-identified non-EUR ancestry could confound dietary habits.

Prior to phenotype derivation, we removed individuals who were pregnant, had kidney disease as defined by ICD10 codes, or a cancer diagnosis within the last year (field 40005). The UKB FFQ consists of quantitative continuous variables (i.e., field 1289, tablespoons of cooked vegetables per day), ordinal non-quantitative variables depending on overall daily/weekly frequency (i.e., field 1329, overall oily fish intake), food types (i.e., milk, spread, bread, cereal, or coffee), or foods never eaten (field 6144, dairy, eggs, sugar, and wheat). Supplementary Data 1 provides a list of UKB fields relating to the corresponding FFQ question for each dietary habit, which can be looked up in the UKB Data Showcase (http://biobank.ndph.ox.ac.uk/showcase/). Ordinal variables were ranked and set to quantitative values, while food types or foods never eaten were converted into a series of binary variables. Variables relating to alcoholic drinks per month were derived from a conglomeration of drinks per month and drinks per week questions answered by different individuals depending

on their response to overall alcohol frequency (field 1558). All 85 single FI dietary phenotypes were then adjusted for age in months and sex, followed by inverse rank normal transformation on continuous FI-QTs. For individuals with repeated FFQ responses, both the dietary variable and the age in months covariate were averaged over all repeated measures. PCs were then derived from all 85 FI-QTs after filling in missing data with the median using the prcomp base function in R. FI-QTs with percent contribution (squared coordinates) greater than expected under a uniform distribution [$1/85 \times 100 = 1.18\%$] were included in Fig. 1 and Supplementary Fig. 4, created using ComplexHeatmap package in R[47]. Phenotype correlation between all 170 dietary habits was estimated using Pearson's pair-wise correlation on complete observations in R. All correlations (phenotypic and genetic) with $P > 0.05/85$ were set to 0. The significance threshold here was selected based on a Bonferroni correction for 85 total FI-QTs to maintain stringency for multiple testing and consistency across phenotype and genetic correlation analyses, while allowing for the nested and non-independent nature of the FFQ questions and derived FI-QTs.

**Heritability, GWAS, and genetic correlation analyses.** Measures of heritability were obtained from BOLT-lmm software (v.2.3.2)[48,49] pseudo-heritability measurement representing the fraction of phenotypic variance explained by the estimated relatedness matrix[48–50]. These estimates were highly correlated with $h_g^2$ estimates using LDSC[51] ($r = 0.988$; Supplementary Fig. 11). BOLT-lmm was unable to calculate $h_g^2$ for "spread type: block margarine vs. never" yielding an "invalid estimate" error, likely indicating no genetic component as BOLT-lmm reported $h_g^2 = 0$ for the highly related "spread type: block margarine vs. any other" dietary habit with ten times the sample size. $Z$-score calculations were used to test for significant differences in heritability between PC1 and its 19 contributing QTs (Eq. 1):

$$\left( Z = \frac{\left( h_{gPC1}^2 - h_{gQT}^2 \right)}{\sqrt{SE - PC1^2 + SE - QT^2}} \right). \tag{1}$$

GWAS of all variables was conducted using the BOLT-lmm software (v.2.3.2)[48,49] linear mixed model association testing to account for relatedness. Additional covariates included in BOLT-lmm analysis for both heritability and GWAS included genotyping array and the first 10 genetic PCs derived on the subset of unrelated EUR individuals using FlashPCA2[52], followed by projection of related individuals on to the PC space. The number of associated loci was determined by clumping of signals within 500 kb windows. Variance explained for each SNP was calculated from the effect size ($\beta$) and effect allele frequency ($f$) as follows: $\beta^2 \times (1 - f) \times 2f$. Pair-wise $r_g$ of all significantly heritable dietary habits were estimated using LDSC[53]. Again, all correlations (phenotypic and genetic) with $P > 0.05/85$ were set to 0. Index SNP annotations were evaluated using Neale Lab UKB GWAS consequence annotations based on Ensembl's Variant Effect Predictor[54]. We performed DEPICT gene set and tissue enrichment analysis, which relies on reconstituted gene sets from publicly available gene sets and pathways and gene expression data to evaluate enrichment in PC1 GWAS loci[55].

Ninety-five percent credible sets were estimated for all 2515 genome-wide significant associations using posterior probabilities as calculated by FINEMAP v1.3[56] using shotgun stochastic search in 500 kb windows. We then used LDstore v1.1[57] to calculate LD and identify SNPs in high LD ($r^2 \geq 0.80$) with any of the 77,229 95% credible set SNPs. The GWAS catalog[18] (downloaded on 13 November 2018) and Neale Lab GWAS v2 conducted in up to 361,194 unrelated individuals (August 2018 release; http://www.nealelab.is/uk-biobank/) was then searched for this list of 339,832 unique SNPs in either any of the 95% credible sets or in high LD with any SNP in the 95% credible sets for any of the 2515 significant GWAS signals in 814 independent loci. We considered any locus as being previously reported if any SNP in its 95% credible set or in high LD ($r^2 \geq 0.80$) with any SNP in the 95% credible SNP was associated with any trait at genome-wide significance ($P < 5.0 \times 10^{-8}$). LocusZoom plots were made using the stand-alone package[58] to include LD information from UKB as determined by LDstore and 95% credible sets from FINEMAP[56,57].

LDSC $r_g$ was again performed on 143 significantly heritable dietary habits with 4336 Neale Lab UKB GWAS traits, 3219 of which had at least one non-missing $r_g$ estimate from LDSC. Using a strict Bonferroni correction threshold for all pair-wise tests between 143 dietary habits and 3219 highly correlated and even overlapping Neale Lab GWAS traits ($P < 0.05/460,317 = 1.09 \times 10^{-7}$), we set all nonsignificant $r_g$ to 0. Supplementary Data 5 represents a complete pair-wise $r_g$ matrix.

Enrichment for olfactory receptor genes among dietary habits was evaluated using 1000 sets of null matched SNPs based on MAF, number of SNPs in LD at various LD thresholds, distance to nearest gene, and gene density using the SNPsnap webtool[59]. We used Fisher's test for enrichment of olfactory receptor genes using SNPsnap's nearest gene annotation for 647 dietary habit GWAS index SNPs and for 1000 sets of null matched SNPs (167 of our index SNPs were excluded for being in the human leukocyte antigen region or had insufficient matches). We based the enrichment analysis contingency table on 842 annotated olfactory receptor genes among 48,903 genes in SNPsnap. Of the 1000 null enrichment analyses, 419 had a Fisher's test estimate equal to or greater than our real data's enrichment estimate.

**Mendelian randomization.** Bidirectional MR was conducted using genome-wide significant index SNPs clumped by 500 kb windows from GWAS in UKB on PC1 ($M = 140$), fluid intelligence scores ($M = 184$), educational attainment ($M = 309$), and BMI ($M = 1165$). Fluid intelligence scores for GWAS in EUR ($N = 232,601$) was derived from both in person and online cognitive tests. Assessment center fluid intelligence scores (field 20016) were averaged for up to three visits and adjusted for average age in months, sex, and assessment center. Online fluid intelligence scores (field 20191) were adjusted for age in months, sex, and townsend deprivation index. The final fluid intelligence score was first set to average assessment center fluid intelligence score, and when missing was filled in with the online fluid intelligence score, for which the combination of these scores were then adjusted for collection method, followed by inverse normal transformation. Educational attainment for GWAS in EUR ($N = 450,884$) was derived using a previously published method based on mapping UKB qualifications field 6138 to US years of schooling[60], followed by adjusted for age in months, sex, and assessment center, and inverse normal transformation. BMI was calculated from weight (field 21002) and standing height (field 50) and averaged from up to three assessment center visits. Average BMI was adjusted for average age in months, average age in months squared, assessment center, and average measurement year, followed by inverse normal transformation conducted in males and females separately. The combined male and female BMI $Z$-scores were then used together for genetic association testing. All GWAS were run in BOLT-lmm adjusted for 10 genetic PCs (calculation described above) and genotyping array.

Genetic instruments for each of the three traits consisted of the complete set of index SNPs for each independent genome-wide significant GWAS locus defined by clumping signals in 500 kb windows. In addition to IVW MR, we also conducted MR Egger to detect pleiotropic effects and WM MR to allow for the inclusion of up to 50% invalid genetic instruments. To test the robustness of any causal association, we repeated MR with filtered genetic instruments using Steiger filtering to remove variants likely influenced by reverse causation and Cook's distance filtering to remove outlying heterogeneous variants[61]. First-pass bidirectional MR included all genome-wide significant SNPs using IVW, MR Egger, and WM using the MendelianRandomization R package[62]. MR sensitivity analysis was conducted by first Steiger filtering to remove variants that explained more variance in the outcome than the exposure as determined by the get_r_from_pn command in TwoSampleMR R package[61,63] and then by filtering out Cook's distance outliers using base R functions. IVW, MR Egger, and WM were then repeated on the filtered genetic instruments. Validation of educational attainment MR results were conducted in two ways. First, a completely external GWAS dataset was used for which 74 discovery stage (not including UKB) genome-wide significant index SNP summary statistics were published online[26]. After removing missing and ambiguous SNPs, the external genetic instrument contained 66 variants. Second, we split the UKB sample into two subsets and conducted GWAS for PC1 and educational attainment as described above in each subset: 1/3 ($N_{pc1} = 149,212$ and $N_{ed} = 150,884$) of the EUR sample was used for genetic instrument variable identification and 2/3 ($N = 300,000$) of the EUR sample was used for testing. MR of the tea intake SNP rs1453548 was conducted using the Wald ratio estimate as calculated by the MendelianRandomization R package[62] using effect size and standard errors from our "cups of tea per day" GWAS and effect size and standard errors from 4358 traits in the Neale Lab GWAS.

**Reporting summary.** Further information on research design is available in the Nature Research Reporting Summary linked to this article.

## Data availability

All 170 derived dietary habits will be returned and shared through UK Biobank (http://biobank.ndph.ox.ac.uk/showcase/) and all GWAS results for the 143 significantly heritable dietary habits are publically available on the Type 2 Diabetes Knowledge Portal (http://www.kp4cd.org/dataset_downloads/t2d).

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

## Acknowledgements

This work was supported by NIH NIDDK 5T32DK110919-02 (J.B.C.), American Diabetes Association 1-19-PDF-028 (J.B.C.) and NIH NIDDK R01DK075787 (J.N.H.). Work was conducted under UK Biobank application 11898.

## Author contributions

J.B.C. conceptualized, analyzed, and interpreted the data and wrote the manuscript. J.N.H. and J.C.F. conceptualized and supervised the analysis and revised the manuscript.

## Competing interests

The authors declare no competing interests.
