## [Peer Review File · Nature Communications]

Reviewers' Comments:

Reviewer #2:

Remarks to the Author:

Review

This manuscript uses the UK Biobank resource to examine diet in up to 450k participants of European ancestry. The genetic analysis uses both single items (referred to as single food intake quantitative traits, FI-QT) and latent variables (termed principal component derived dietary patterns, PC-DP) to examine diet. A total of 85 FI-QTs are assessed along with 85 PC-DPs. The main results are that the majority of the 85 FI-QTs and the 85 PC-DPs are heritable and show genetic correlations with other traits, and that 814 loci were identified as being linked to various aspects of diet.

Furthermore an analysis of the first PC indicated links between diet and intelligence and education with significant SNPs as well as biological pathways appearing to be shared between these three phenotypes. Mendelian Randomisation indicated a bi-directional causal relationship between intelligence and education with diet.

I think this is a very valuable manuscript and should be seen as the gold standard in what to expect from a GWAS today. Its methodological contributions include using a latent variable approach rather than simply summing the scores from pre-selected subsets of items, the use of linear mixed models rather than simple regression. Furthermore, it contains some of the best Figures I've seen for summarising a large amount of data. Figure 1 is particularly impressive as a way of communicating where the variance found in PC1 is drawn from, as well as the phenotypic and genetic relationship between PC1 and the items that load most strongly on it. It really is a nice piece of work.

However I have a number of concerns regarding the methods used particularly the phenotyping as well as the resulting interpretation by the authors.

1. For the Mendelian randomisation the fluid intelligence test score has been adjusted for the Townsend Score. As adjusting for heritable covariates can bias GWAS estimates(1) I would advise this analysis to be repeated without controlling for the Townsend Score.

2. Including 10 PCs in the LMM seems low. The LDSC regression intercepts should be made available too to ascertain if there is any residual stratification.

3. Too often statements are made that compare two estimates of heritability but no formal test is ever used to test if this difference is significant.

4. The caption for Supplementary Figure 5A and Supplementary Figure 5B has been swapped. The finding presented here is, I think, quite interesting as it is saying that more heritable phenotypes have SNPs with larger effect sizes (looking at those phenotypes with a sample size of >400k) rather than simply a greater number of causal SNPs. However, as there is a lot of dependency across the phenotypes it would be better to have these Figures made again but only including the PC-DPs. This would allow you to see if this trend exists across the most independent data points.

5. Furthermore, the large r_g between PC1 with educational attainment, and the overlap in loci identified with intelligence, indicates to me that some of what makes up choice of diet in the UK today is linked to socioeconomic status. Previous studies have shown that SES is heritable (2) as well as that intelligence is causally linked to difference in SES in UK Biobank(3). I think that the authors should discuss this link as it may indicate that PC1 is tapping both variance associated with obesity related traits, as evidenced by the 38 loci associated with obesity traits, as well as drivers of SES, as evidenced by the 24 loci associated with PC1 and with intelligence and education.

6. Looking at supplementary table 5 it seems that PC1 has no genetic correlation with BMI,

multiple measures of adiposity, diabetes, heart disease. What do the authors think about this? Shouldn't measures of diet be linked to disease with a large dietary component?

Minor points

1. The authors state "The relationships between the 83 FI-QTs and the 60 PC-DPs that have significant heritability are illustrated in Supplementary Figure 4." However, no relationship between these variables is plotted, only the heritability of each. It is a valuable figure but this caption is wrong. Also why are 60 used here? It was only 20 that were selected on the basis of their eigenvalues being greater than 1.
2. Regarding the Mendelian randomisation analysis between BMI and PC1. Wouldn't you need a significant rg between these two phenotypes?

References

1. Aschard H, Vilhjálmsson Bjarni J, Joshi Amit D, Price Alkes L, Kraft P (2015): Adjusting for Heritable Covariates Can Bias Effect Estimates in Genome-Wide Association Studies. *The American Journal of Human Genetics*. 96:329-339.
2. Hill WD, Hagenaars SP, Marioni RE, Harris SE, Liewald DC, Davies G, et al. (2016): Molecular genetic contributions to social deprivation and household income in UK Biobank. *Current Biology*. 26:3083-3089.
3. Hill WD, Davies NM, Ritchie SJ, Skene NG, Bryois J, Bell S, et al. (2019): Genetic analysis identifies molecular systems and biological pathways associated with household income. [bioRxiv.573691](https://doi.org/10.1101/573691).

Reviewer #3:

Remarks to the Author:

The authors report the results of a GWAS analysis of dietary habits, using data from UK Biobank. This is supplemented by Mendelian randomization analyses to identify potential causal pathways to and from dietary phenotypes.

This is an extremely ambitious study, given the difficulties associated with capturing the diversity and complexity of dietary behaviour, that attempts to answer an extremely important research question. The authors use principal component analysis in an attempt to resolve this complexity, which is a strength of the study. Much of the study turns on the results relating to a single principal component, which in turn appears to be strongly influenced by a single item (type of bread consumed), which the authors describe as a Western vs "prudent" diet.

I have two main concerns, which principally relate to the interpretation of results rather than the methods per se. One is the Food Frequency (FFQ) data that form the basis of the phenotype, and the other is potential structure within UK Biobank data.

FFQ data are known to be noisy / unreliable measures of actual dietary intake, although they are still widely used given their ease of administration. However, this means that interpreting the results of a GWAS is complicated because the questionnaires may be picking up other phenotypes (e.g., memory, the tendency to self-report in a favourable way, etc.) than the intended phenotype. For this reason, interpreting the results of the GWAS results is challenging, and assuming that the GWAS-significant hits necessarily reflect underlying biology (see Gage et al., 2016) may be problematic.

This issue is compounded by the fact that there is latent structure within the UK Biobank genetic

data that reflects geographical structure (and by extension socioeconomic confounding) (Haworth et al., 2019). This can induce bias in observed associations, even in Mendelian randomization analyses. The fact that one of the main drivers of the primary principal component of interest – type of bread consumed – is so strong socially patterned within the UK raises the possibility that the genetic variants identified are also capturing socioeconomic confounding (I strongly suspect this phenotype will be patterned by region).

The paper might be strongly focused on developing a tool that captures variation in dietary behaviour (although with the caveats associated with my first point) and focusing on using this to understand causal pathways to and from dietary behaviour. The relative lack of clear evidence of a causal pathway from diet to relevant health outcomes is a cause for concern however, and reinforces the point that the data may be capturing other phenotypes than simply dietary behaviour. Other approaches are possible within UK Biobank (e.g. Davies et al., 2018), but this would be quite a different study.

I sign my reviews.

Marcus Munafò

References

Davies et al. (2018). The causal effects of education on health outcomes in the UK Biobank. *Nature Human Behaviour*, 2, 117-125.

Gage et al. (2016) G=E: What GWAS can tell us about the environment. *PLOS Genetics*, 12, e1005765.

Haworth et al. (2019). Apparent latent structure within the UK Biobank sample has implications for epidemiological analysis. *Nature Communications*, 10, 333.

NCOMMS-19-17863-T

Response to reviews for “Comprehensive genomic analysis of dietary habits in UK Biobank identifies hundreds of genetic loci and establishes causal relationships between educational attainment and healthy eating”

We would like to thank the reviewers for their constructive comments.

Our primary extensive change to the manuscript relates to additional analysis testing for the effects of residual confounding in our dietary habit GWAS. We have added the following analyses and text to the manuscript, and refer to this paragraph as the “primary additional analysis paragraph” throughout the review. We additionally include all areas in which we made changes to the manuscript text, *and italicize specific changes*.

“Recent work in a non-mixed model setting has demonstrated the presence of latent structure in UK Biobank co-incident with outcomes such as educational attainment, which is also associated with dietary intake (Haworth et al. and Shimakawa et al). We found that our linear mixed model dietary habit GWAS had little to no residual confounding as estimated by LD score regression intercepts (median = 1.018, interquartile range (IQR):1.011-1.030) and ratios (median = 0.087, IQR: 0.059-0.0123; Supplementary Table 1). Furthermore, large-scale sensitivity analysis re-conducting all 143 significantly heritable dietary habit GWAS additionally adjusting for assessment center in the genetic model only found slight attenuation of heritability (median from 4.6% to 4.2%, IQR from 2.4-7.9% to 2.2-7.4%). We report the results in this paper without additional adjustment for assessment center.”

Point-by-point responses:

Reviewer #2

- 1. For the Mendelian randomisation the fluid intelligence test score has been adjusted for the Townsend Score. As adjusting for heritable covariates can bias GWAS estimates(1) I would advise this analysis to be repeated without controlling for the Townsend Score.**

We agree that an initial analysis should avoid the adjustment of heritable covariates, which may bias results when the genetic variant is associated with the covariate (Townsend deprivation index) either alone or independently of the outcome (intelligence). We chose to initially correct for both assessment center and Townsend deprivation index in the fluid intelligence GWAS to recapitulate the genetic model used in the most recent intelligence GWAS publication (Sniekers et al 2017).

The Townsend deprivation index in UK Biobank (field 189) is based on national census data and was used as an additional proxy for geographic location in addition to assessment center. As highlighted in the primary additional analysis paragraph above, we find minimal residual population stratification. This suggests to us that while assessment center and Townsend deprivation index are both significantly associated with several outcomes (including both dietary habits and fluid intelligence scores) and are co-incident with geographical genetic variation (Haworth et al 2019), linear mixed models, the gold standard GWAS analytical method,

sufficiently captures residual population structure in our analyses of dietary habits in UK Biobank to avoid substantial confounding.

Nevertheless, we agree with the value in performing the suggested sensitivity analysis and demonstrate below highly concordant results with and without adjustment for Townsend deprivation and assessment center on fluid intelligence test score heritability, GWAS, and Mendelian randomization with PC1.

	Original INT results (adjustment)	Sensitivity analysis INT results (no adjustment)
SNP Heritability (h²_g)	26.7%	26.9%
Genome-wide significant SNPs	184	175
Weighted Median Mendelian Randomization Exposure: INT Outcome: PC1	Estimate = 0.261 P-value = 7.71×10^{-13}	Estimate = 0.278 P-value = 1.15×10^{-13}
Weighted Median Mendelian Randomization Exposure: PC1 Outcome: INT	Estimate = 0.074 P-value = 8.04×10^{-7}	Estimate = 0.110 P-value = 1.10×10^{-12}

2. Including 10 PCs in the LMM seems low.

We believe that including PCs on top of a linear mixed model is fairly uncommon and rather conservative. However, we chose to include 10 PCs that were re-derived from our European only sample (as opposed to using the multi-ancestry PCs provided by UKB) as an added precautionary measure and to speed up the mixed model computation. As depicted below in our PC bi-plots, our lower order PCs capture little to no additional variation.

We also include below a fish plot comparing significance and directionality of dietary pattern PC1 LMM GWAS results with 10 PCs vs. 40 PCs. P-values are signed based on consistency in the direction of the betas between the two comparison models. Thus quadrant 1 represents $-\log(P\text{-values})$ of SNPs with effect sizes in the same direction and quadrant 3 represents $-\log(P\text{-values})$ of SNPs with effect sizes in opposite directions. The vast majority of SNPs fall very close to the line of identity in quadrant 1, and thus there is minimal residual confounding not captured by the linear mixed model and 10 PCs.

FISH Plot - PC1 - BOLTImm with 10PCs vs 40PCs

-- The LDSC regression intercepts should be made available too to ascertain if there is any residual stratification.

LD score regression (LDSC) intercepts and ratios are now included in Table S1 and the inclusion of this analysis is described in the primary additional analysis paragraph above. The 6-number descriptive summary of the LDSC intercepts and ratios for our heritable traits are summarized below, essentially demonstrating little to no residual confounding that is not controlled for with our linear mixed model as the intercepts are near one and the bulk of the ratios are near zero (Bulik-Sullivan et al 2015). Only PC62 had a ratio greater than 40%, and only five dietary habits had ratios greater than 30% indicating mild residual confounding, including PC40, PC44, PC48, PC62, and never eat wheat vs. no wheat restrictions.

	Min	Q1	Median	Mean	Q3	Max
LDSC intercept	0.993	1.011	1.018	1.021	1.030	1.074
LDSC ratio	0.00	0.059	0.087	0.102	0.123	0.599

3. Too often statements are made that compare two estimates of heritability but no formal test is ever used to test if this difference is significant.

We have added formal tests comparing heritability estimates between PC1 and its 19 contributing QTs and updated the text as follows:

“The first PC-DP (hereafter referred to as PC1) is among the most heritable dietary patterns (PC1 $h_g^2=13.6\%$, Table 1), and is more heritable than all of its individual contributing FI-QTs (*all h_g^2 comparisons have $P < 5.0 \times 10^{-45}$*).”

We have added the following sentence to the Methods

“Z-score calculations were used to test for significant differences in heritability between PC1 and its 19 contributing QTs ($Z = \frac{(h_{g-PC1}^2 - h_{g-QT}^2)}{\sqrt{SE-PC1^2 + SE-QT^2}}$)”

4. The caption for Supplementary Figure 5A and Supplementary Figure 5B has been swapped.

We have updated Figure 5 to correctly align with the caption and appreciate the reviewer bringing this to our attention.

--The finding presented here is, I think, quite interesting as it is saying that more heritable phenotypes have SNPs with larger effect sizes (looking at those phenotypes with a sample size of >400k) rather than simply a greater number of causal SNPs. However, as there is a lot of dependency across the phenotypes it would be better to have these Figures made again but only including the PC-DPs. This would allow you to see if this trend exists across the most independent data points.

We have added 2 panels to Figure S5 (below). Similar to Figure 2, these plots depict effect size and p-value of the “best” SNPs for each trait plotted by either its total number of genome-wide significant loci (A) or total variance explained by the genome-wide significant loci (B), though as the reviewer suggested, restricted to independent dietary habits, PC-DPs only. The trend does persist among phenotypically independent dietary patterns.

We have also updated the text as follows:

“Across the 143 dietary traits, there was a clear positive correlation of heritability estimates (h_g^2) with the number of significant loci and the variance explained by these loci (Figure 2), a pattern that persisted when limited to phenotypically independent DP-PCs, with outliers often explained by smaller sample size (Supplementary Figure 5).”

5. Furthermore, the large r_g between PC1 with educational attainment, and the overlap in loci identified with intelligence, indicates to me that some of what makes up choice of diet in the UK today is linked to socioeconomic status. Previous studies have shown that SES is heritable (2) as well as that intelligence is causally linked to difference in SES in UK Biobank(3). I think that the authors should discuss this link as it may indicate that PC1 is tapping both variance associated with obesity related traits, as evidenced by the 38 loci associated with obesity traits, as well as drivers of SES, as evidenced by the 24 loci associated with PC1 and with intelligence and education.

We have including some key phrases and genetic correlations to the following sections in the discussion to emphasize the overlap between PC1 and educational attainment and BMI, and between educational attainment and income and intelligence.

“While bidirectional MR demonstrates some pleiotropic effects between educational attainment and PC1 *indicating either shared biology or upstream common cause(s)*, the relative strengths of these causal estimates suggests that higher educational attainment and/or correlated phenotypes (such as socioeconomic status [$r_g = 0.77$] or factors related to school performance, *such as fluid intelligence test scores* [$r_g = 0.68$]) *either directly or indirectly* shift eating habits towards a healthier, more prudent diet.”

“Of note, the interpretation of the genetic component of predominantly environmental traits, such as dietary intake, can be complicated. As we have shown, dietary habits are highly correlated both with each other and with non-dietary traits, suggesting that any single dietary phenotype may represent a broader diet and lifestyle. For instance, dietary pattern PC1 with both high genetic correlation with educational attainment and large overlap in genome-wide significant loci with BMI, is capturing variance in both obesity-related traits and measures of socioeconomic status.”

We have also highlighted this in the results section when assessing overlap of top genetic signals as follows:

“Of these 83, 24 were associated with intelligence or educational attainment and not obesity-related traits; 38 were associated with obesity-related traits and not intelligence or educational attainment, indicating PC1 may be responsive to distinct sets of heritable factors related separately to educational attainment and to BMI.”

6. Looking at supplementary table 5 it seems that PC1 has no genetic correlation with BMI, multiple measures of adiposity, diabetes, heart disease. What do the authors think about this? Shouldn't measures of diet be linked to disease with a large dietary component?

A closer look at the genetic correlation between PC1 and these traits revealed nominally significant, albeit small genetic correlations. Specifically PC1 is genetically correlated with BMI (field 21001_irnt: $r_g = -0.0563$, $P = 0.0243$), diabetes (self-reported any diabetes field 2443: $r_g = -0.0786$, $P = 0.0013$), and 'major coronary heart disease event' (field I9 CHD: $r_g = -0.1809$, $P = 2.27 \times 10^{-7}$). The genetic correlation matrix in Table S5 uses a strict Bonferroni-adjusted significance threshold, reporting any non-significant genetic correlations with $P > 1.09 \times 10^{-7}$ as 0.

Interestingly, a large number of SNPs that are genome-wide significant for PC1 are at least nominally significant for BMI, T2D from DIAGRAM, CAD from CARDIOGRAM

Genome-wide significant SNPs from	Lookup GWAS	Number of SNPs present in lookup GWAS	$P < 0.05$ in lookup GWAS	$P < 5 \times 10^{-8}$ in lookup GWAS
PC1	BMI	140	78	29
PC1	T2D	94	18	2
PC1	CAD	105	12	1
BMI	PC1	1165	337	18
T2D	PC1	37	3	1
CAD	PC1	38	10	2

This potentially indicates that while there is a nontrivial overlap between the top SNPs affecting PC1 and BMI and cardiometabolic disease, there is a relatively low overall genetic correlation among all SNPs. This could indicate the presence of shared genetic pathways that have large effects on both PC1 and BMI and cardiometabolic disease and the presence of non-overlapping

pathways specific to either PC1 or BMI and cardiometabolic disease that therefore lower the overall genetic correlation between the two respective traits.

Another key aspect to this analysis is that our dietary habits are non-isocaloric, indicating that they were not adjusted for total energy intake. As Reviewer #2 previously mentioned, adjusting for a heritable covariate leads to a series of different biases depending on the precise relationships between the genetic variants, the outcome, and the heritable covariate. However, it is important to point out that individuals who consume a large total energy intake will in turn eat more of any given food, and that may be due to a larger required energy load which is influenced by BMI and physical activity. Dietary habits that are adjusted for total energy intake might have a different, perhaps stronger genetic correlation with BMI and cardiometabolic disease. Though we indicate in the manuscript that single foods and dietary patterns display little to no correlation with total energy intake in UK Biobank, we plan to explore how BMI and physical activity affect the relationships between genetics and dietary intake in a more depth future analysis.

We appreciate the attention brought to this and have expanded on this in the follow sections of the manuscript

“In contrast to the educational attainment analyses, we were unable to provide robust evidence of a causal relationship in either direction between BMI and PC1 due to significant heterogeneous pleiotropic effects leading to inconsistent causal effect estimates (Table 2, Supplementary Figure 8). Taken together with an overlap between genome-wide significant variants and small overall genetic correlation ($r_g = -0.056$, $P = 0.024$), we postulate that BMI and our non-isocaloric PC1 have a highly complex multi-pathway relationship for which there may be overlapping genetic etiology, but a mostly-distinct underlying genetic architecture.”

Minor points

1. The authors state “The relationships between the 83 FI-QTs and the 60 PC-DPs that have significant heritability are illustrated in Supplementary Figure 4.” However, no relationship between these variables is plotted, only the heritability of each. It is a valuable figure but this caption is wrong.

We agree with the reviewer and have corrected the text to say:

“Phenotypic and genetic correlation between the 60 PC-DPs and their significantly contributing FI-QTs are illustrated in Supplementary Figure 4.”

--Also why are 60 used here? It was only 20 that were selected on the basis of their eigenvalues being greater than 1.

We filtered dietary habits for downstream genetic analysis based on genetic variance explained (SNP heritability) instead of phenotypic variance explained (based on eigenvalues). While we thought it was interesting to mention the number of PCs that had eigenvalues greater than 1, we used heritability instead as it could be applied to both quantitative traits and principal components and genetic variance was more appropriate for our downstream genetic analyses.

We have clarified this in the text as follows:

“Overall, 84.1% of dietary habits analyzed (83/85 FI-QTs and 60/85 PC-DPs) were significantly heritable (as assessed by h_g^2 $P < 0.05 / 170 = 2.9 \times 10^{-4}$; see Methods, Table 1, Supplementary Table 1, and Supplementary Figure 3), displayed extensive genetic correlation (r_g ; Supplementary Table 3), and were included in downstream genomic analysis.”

2. Regarding the Mendelian randomisation analysis between BMI and PC1. Wouldn't you need a significant r_g between these two phenotypes?

Yes, in theory there needs to be genetic overlap between the two traits. Traditional MR analyses, as we have done here, only use the top genetic variants as the genetic instrument for the exposure. As such, we may be only capturing one aspect of PC1 biology that overlaps with top BMI genetic variants. Our MR results suggest high heterogeneity and directional pleiotropy, which could be related to the contrast seen between the strong genetic overlap between top SNPs vs low overall shared genetic architecture (i.e. genetic correlation), again consistent with the potential for complex shared and unique pathways between BMI and PC1. Future analyses using both clustered and larger sets of SNPs as genetic instruments could help tease apart these relationships, though we believe this beyond the scope of this manuscript. Please see the changes to the manuscript in Reviewer #2 comment 6 above.

Reviewer #2 References

1. Aschard H, Vilhjálmsdóttir Bjarni J, Joshi Amit D, Price Alkes L, Kraft P (2015): Adjusting for Heritable Covariates Can Bias Effect Estimates in Genome-Wide Association Studies. *The American Journal of Human Genetics*. 96:329-339.
2. Hill WD, Hagenaars SP, Marioni RE, Harris SE, Liewald DC, Davies G, et al. (2016): Molecular genetic contributions to social deprivation and household income in UK Biobank. *Current Biology*. 26:3083-3089.
3. Hill WD, Davies NM, Ritchie SJ, Skene NG, Bryois J, Bell S, et al. (2019): Genetic analysis identifies molecular systems and biological pathways associated with household income. *bioRxiv*.573691.

Reviewer #3

1. FFQ data are known to be noisy / unreliable measures of actual dietary intake, although they are still widely used given their ease of administration. However, this means that interpreting the results of a GWAS is complicated because the questionnaires may be picking up other phenotypes (e.g., memory, the tendency to self-report in a favourable way, etc.) than the intended phenotype. For this reason, interpreting the results of the GWAS results is challenging, and assuming that the GWAS-significant hits necessarily reflect underlying biology (see Gage et al., 2016) may be problematic.

We wholeheartedly agree that interpretation of GWAS findings with lifestyle factors is complex. As the reviewer mentions, questionnaire responses may capture memory or social desirability, while dietary intake itself is correlated with what may be hundreds of other traits, including

physical activity, BMI, socioeconomic status, disease status, geographical location, climate, employment, etc. Even “simple” case-control disease phenotypes may suffer from biases relating to self-reported diagnosis, lifestyle intervention, and pre-clinical disease status. Environmental traits are nearly impossible to capture with certainty, and thus we must rely on studies that use surrogate best-available measures while presenting these limitations. We appreciate this point and have emphasized this throughout the revision to our manuscript as follows:

- 1) *“Of note, the interpretation of the genetic component of predominantly environmental traits, such as dietary intake, can be complicated. As we have shown, dietary habits are highly correlated both with each other and with non-dietary traits, suggesting that any single dietary phenotype may represent a broader diet and lifestyle. For instance, dietary pattern PC1 with both high genetic correlation with educational attainment and large overlap in genome-wide significant loci with BMI, is capturing variance in both obesity-related traits and measures of socioeconomic status. Additionally, similar to most nutritional epidemiology studies, measures of dietary intake in UKB are based on self-reported questionnaire data, which intrinsically suffers biases based on memory and favorable reporting (Smith 1991, Hebert 1995), further complicating the interpretation of genomic results.”*
- 2) *“The large and broad overlap in both the significant loci and the overall genetic make-up of our dietary habits with other traits, many of which do not have well-established biological links with diet, emphasizes a need for exploring correlation vs. causation. ... Therefore, we sought to understand the cause-and-effect relationships and degree of pleiotropy between PC1 and educational attainment, fluid intelligence scores, and BMI using bidirectional MR.” (Davey Smith & Hemani 2014)*
- 3) *“Importantly, because the instrumental variables used for educational attainment are not mechanistically linked directly to educational attainment/intelligence, it remains possible that causal influences on PC1 could be due to unmeasured heritable factor(s) that are themselves causal for educational attainment/intelligence. Furthermore, because PC1 is highly correlated with hundreds of additional outcomes, educational attainment (or traits correlated with educational attainment) could be influencing a larger complex phenotype made of many lifestyle factors that is captured here by PC1 dietary preferences.”*
- 4) *“To test whether the “prudent” PC1 dietary pattern is capturing a lifestyle that is likely to causally influence disease risk....”*
- 5) *“While bidirectional MR demonstrates some pleiotropic effects between educational attainment and PC1 indicating either shared biology or upstream common cause(s), the relative strengths of these causal estimates suggests that higher educational attainment and/or correlated phenotypes (such as socioeconomic status [$r_g = 0.77$] or factors related to school performance, such as fluid intelligence test scores [$r_g = 0.68$]) either directly or indirectly shift eating habits towards a healthier, more prudent diet.”*

--This issue is compounded by the fact that there is latent structure within the UK Biobank genetic data that reflects geographical structure (and by extension socioeconomic confounding) (Haworth et al., 2019). This can induce bias in observed associations, even in Mendelian randomization analyses. The fact that one of the main drivers of the primary principal component of interest – type of bread consumed – is so strong socially patterned within the UK raises the possibility that the genetic variants

identified are also capturing socioeconomic confounding (I strong suspect this phenotype will be patterned by region).

We believe that our analysis using linear mixed models minimizes the effects of residual population structure in UK Biobank not captured using traditional simple regression approaches with genetic PCs alone. We have formally tested this by including LDSC intercepts and ratios which demonstrate little to no residual confounding (see the primary additional analysis paragraph and response to Reviewer #2 comment #2 above). We have additionally conducted a sensitivity analysis on all 143 significantly heritable dietary habits by including assessment center in the linear mixed model as a covariate and found only a slight attenuation of genomic signal (see primary additional analysis paragraph above and PC1-specific results in table below):

	Original PC1 results (no center adjustment)	Sensitivity analysis PC1 results (center adjustment)
SNP Heritability (h²_g)	13.5%	13.0%
Genome-wide significant SNPs	140	126
Weighted Median Mendelian Randomization Exposure: SES Outcome: PC1	Estimate = 0.823 P-value = 6.59×10^{-113}	Estimate = 0.807 P-value = 2.30×10^{-110}
Weighted Median Mendelian Randomization Exposure: PC1 Outcome: SES	Estimate = 0.199 P-value = 7.44×10^{-56}	Estimate = 0.194 P-value = 1.98×10^{-49}

We added the following text and results to the manuscript:

“Results were also in high agreement with a repeated MR analysis using our sensitivity analysis PC1 GWAS results adjusted for assessment center (Supplementary Table 7).”

Furthermore, we present our results between PC1 and educational attainment in light of significant bidirectional MR effects, indicating that PC1 is in fact capturing some of the same biology that pertains to traits correlated with educational attainment. We have revised our interpretation of results to clarify this key point as laid out in previous responses to reviewer comments above.

- 2. The paper might be strong focused on developing a tool that captures variation in dietary behaviour (although with the caveats associated with my first point) and focusing on using this to understand causal pathways to and from dietary behaviour. The relative lack of clear evidence of a causal pathway from diet to relevant health outcomes is a cause for concern however, and reinforces the point that the data may be capturing other phenotypes than simply dietary behaviour. Other approaches are possible within UK Biobank (e.g. Davies et al., 2018), but this would be quite a**

different study.

We agree the one of the most likely causes for a lack of clear evidence from PC1 to cardiometabolic disease is related to the complex lifestyle that PC1 is capturing as a dietary habit. It is predominantly driven by bread type and is highly phenotypically and genetically correlated with hundreds of other outcomes. We have updated the text extensively to clarify this complication to interpretation (highlighted throughout the above reviewer comments), including an additional paragraph in the Discussion section (see reviewer #3, comment #1, update #1). We believe the development of a tool to test the causal relationships between variation in dietary intake and health outcomes is a valuable next step and beyond the scope of this manuscript. We also agree, as indicated by Davies et al, that the years of schooling educational attainment measurement is indirectly capturing other correlated measures that may confound the true effects of education on health outcomes. Our manuscript has been revised to reflect these limitations to interpretation. Those changes are highlighted above throughout this response to reviewer comments. As suggested, an analysis using the educational reform changes in the UK unaffected by genomic confounding (Davies et al 2018), is both an intriguing approach and beyond the scope of this genomic analysis.

Reviewer #3 References

Davies et al. (2018). The causal effects of education on health outcomes in the UK Biobank. *Nature Human Behaviour*, 2, 117-125.

Gage et al. (2016) G=E: What GWAS can tell us about the environment. *PLOS Genetics*, 12, e1005765.

Haworth et al. (2019). Apparent latent structure within the UK Biobank sample has implications for epidemiological analysis. *Nature Communications*, 10, 333.

Response to Reviewer Comments References

Bulik-Sullivan, B. K., Loh, P. R., Finucane, H. K., Ripke, S., Yang, J., Patterson, N., ... & Schizophrenia Working Group of the Psychiatric Genomics Consortium. (2015). LD Score regression distinguishes confounding from polygenicity in genome-wide association studies. *Nature genetics*, 47(3), 291.

Davey Smith, G., & Hemani, G. (2014). Mendelian randomization: genetic anchors for causal inference in epidemiological studies. *Human molecular genetics*, 23(R1), R89-R98.

Davies et al. (2018). The causal effects of education on health outcomes in the UK Biobank. *Nature Human Behaviour*, 2, 117-125.

Haworth, S., Mitchell, R., Corbin, L., Wade, K. H., Dudding, T., Budu-Aggrey, A., ... & Davies, N. (2019). Apparent latent structure within the UK Biobank sample has implications for epidemiological analysis. *Nature communications*, 10(1), 333.

Hebert, J. R., Clemow, L., Pbert, L., Ockene, I. S., & Ockene, J. K. (1995). Social desirability bias in dietary self-report may compromise the validity of dietary intake measures. *International journal of epidemiology*, 24(2), 389-398.

Shimakawa, T., Sorlie, P., Carpenter, M. A., Dennis, B., Tell, G. S., Watson, R., & Williams, O. D. (1994). Dietary intake patterns and sociodemographic factors in the Atherosclerosis Risk in Communities Study. *Preventive medicine*, 23(6), 769-780.

Smith, A. F., Jobe, J. B., & Mingay, D. J. (1991).. *Applied Cognitive Psychology*, 5(3), 269-296.

Sniekers, S., Stringer, S., Watanabe, K., Jansen, P. R., Coleman, J. R., Krapohl, E., ... & Amin, N. (2017). Genome-wide association meta-analysis of 78,308 individuals identifies new loci and genes influencing human intelligence. *Nature genetics*, 49(7), 1107.

Reviewers' Comments:

Reviewer #2:

Remarks to the Author:

NCOMMS-19-17863-T

Response to reviews for "Comprehensive genomic analysis of dietary habits in UK Biobank identifies hundreds of genetic loci and establishes causal relationships between educational attainment and healthy eating"

We would like to thank the reviewers for their constructive comments.

Our primary extensive change to the manuscript relates to additional analysis testing for the effects of residual confounding in our dietary habit GWAS. We have added the following analyses and text to the manuscript, and refer to this paragraph as the "primary additional analysis paragraph" throughout the review. We additionally include all areas in which we made changes to the manuscript text, and italicize specific changes.

"Recent work in a non-mixed model setting has demonstrated the presence of latent structure in UK Biobank co-incident with outcomes such as educational attainment, which is also associated with dietary intake (Haworth et al. and Shimakawa et al). We found that our linear mixed model dietary habit GWAS had little to no residual confounding as estimated by LD score regression intercepts (median = 1.018, inter-quartile range (IQR):1.011-1.030) and ratios (median = 0.087, IQR: 0.059-0.0123; Supplementary Table 1). Furthermore, large-scale sensitivity analysis re-conducting all 143 significantly heritable dietary habit GWAS additionally adjusting for assessment center in the genetic model only found slight attenuation of heritability (median from 4.6% to 4.2%, IQR from 2.4-7.9% to 2.2-7.4%). We report the results in this paper without additional adjustment for assessment center."

Point-by-point responses:

Reviewer #2

For the Mendelian randomisation the fluid intelligence test score has been adjusted for the Townsend Score. As adjusting for heritable covariates can bias GWAS estimates(1) I would advise this analysis to be repeated without controlling for the Townsend Score.

We agree that an initial analysis should avoid the adjustment of heritable covariates, which may bias results when the genetic variant is associated with the covariate (Townsend deprivation index) either alone or independently of the outcome (intelligence). We chose to initially correct for both assessment center and Townsend deprivation index in the fluid intelligence GWAS to recapitulate the genetic model used in the most recent intelligence GWAS publication (Sniekers et al 2017). The Townsend deprivation index in UK Biobank (field 189) is based on national census data and was used as an additional proxy for geographic location in addition to assessment center. As highlighted in the primary additional analysis paragraph above, we find minimal residual population stratification. This suggests to us that while assessment center and Townsend deprivation index are both significantly associated with several outcomes (including both dietary habits and fluid intelligence scores) and are co-incident with geographical genetic variation (Haworth et al 2019), linear mixed models, the gold standard GWAS analytical method, sufficiently captures residual population structure in our analyses of dietary habits in UK Biobank to avoid substantial confounding.

Nevertheless, we agree with the value in performing the suggested sensitivity analysis and demonstrate below highly concordant results with and without adjustment for Townsend deprivation and assessment center on fluid intelligence test score heritability, GWAS, and Mendelian randomization with PC1.

Original INT results (adjustment) Sensitivity analysis INT results (no adjustment)

SNP Heritability (h²_g) 26.7% 26.9%
Genome-wide significant SNPs 184 175
Weighted Median Mendelian Randomization
Exposure: INT
Outcome: PC1 Estimate = 0.261
P-value = 7.71 x 10⁻¹³ Estimate = 0.278
P-value = 1.15 x 10⁻¹³
Weighted Median Mendelian Randomization
Exposure: PC1
Outcome: INT Estimate = 0.074
P-value = 8.04 x 10⁻⁷ Estimate = 0.110
P-value = 1.10 x 10⁻¹²

Referee response.

The Sniekers et al paper is not the most recent GWAS of intelligence as this only includes data from the initial release of the UK Biobank genetic data. Since then the full UK Biobank cognitive data has been published as part of a large GWAS¹ and this has been followed by replication GWASs.^{2, 3}

Furthermore, the authors state that the Townsend score is an additional proxy for geographic location in a similar manner to assessment centre. Whilst the authors are correct that the Townsend score captures geographical genetic variation, this is not due to it capturing population stratification issues, but rather indicates that the heritable traits linked to differences in socioeconomic status are not distributed evenly across the UK.^{4, 5}

However, the sensitivity analysis conducted by the author's shows that these covariates make very little difference to the analyses. It would be better however, if they could only omit the Townsend score rather than the Townsend score and the assessment centre.

2. Including 10 PCs in the LMM seems low.

We believe that including PCs on top of a linear mixed model is fairly uncommon and rather conservative. However, we chose to include 10 PCs that were re-derived from our European only sample (as opposed to using the multi-ancestry PCs provided by UKB) as an added precautionary measure and to speed up the mixed model computation. As depicted below in our PC bi-plots, our lower order PCs capture little to no additional variation.

We also include below a fish plot comparing significance and directionality of dietary pattern PC1 LMM GWAS results with 10 PCs vs. 40 PCs. P-values are signed based on consistency in the direction of the betas between the two comparison models. Thus quadrant 1 represents $-\log(P\text{-values})$ of SNPs with effect sizes in the same direction and quadrant 3 represents $-\log(P\text{-values})$ of SNPs with effect sizes in opposite directions. The vast majority of SNPs fall very close to the line of identity in quadrant 1, and thus there is minimal residual confounding not captured by the linear mixed model and 10 PCs.

-- The LDSC regression intercepts should be made available too to ascertain if there is any residual stratification.

LD score regression (LDSC) intercepts and ratios are now included in Table S1 and the inclusion of this analysis is described in the primary additional analysis paragraph above. The 6-number

descriptive summary of the LDSC intercepts and ratios for our heritable traits are summarized below, essentially demonstrating little to no residual confounding that is not controlled for with our linear mixed model as the intercepts are near one and the bulk of the ratios are near zero (Bulik-Sullivan et al 2015). Only PC62 had a ratio greater than 40%, and only five dietary habits had ratios greater than 30% indicating mild residual confounding, including PC40, PC44, PC48, PC62, and never eat wheat vs. no wheat restrictions.

Min	Q1	Median	Mean	Q3	Max	
LDSC intercept	0.993	1.011	1.018	1.021	1.030	1.074
LDSC ratio	0.00	0.059	0.087	0.102	0.123	0.599

Referee response.

I am very happy with this thorough response.

3. Too often statements are made that compare two estimates of heritability but no formal test is ever used to test if this difference is significant.

We have added formal tests comparing heritability estimates between PC1 and its 19 contributing QTs and updated the text as follows:

"The first PC-DP (hereafter referred to as PC1) is among the most heritable dietary patterns (PC1 $h_g^2=13.6\%$, Table 1), and is more heritable than all of its individual contributing FI-QTs (all h_g^2 comparisons have $P < 5.0 \times 10^{-45}$)."

We have added the following sentence to the Methods

"Z-score calculations were used to test for significant differences in heritability between PC1 and its 19 contributing QTs ($Z = ((h_g(PC1))^2 - h_g(QT))^2) / \sqrt{[SE(PC1)]^2 + [SE(QT)]^2}$)"

Referee response.

The authors have addressed my concerns here.

4. The caption for Supplementary Figure 5A and Supplementary Figure 5B has been swapped.

We have updated Figure 5 to correctly align with the caption and appreciate the reviewer bringing this to our attention.

--The finding presented here is, I think, quite interesting as it is saying that more heritable phenotypes have SNPs with larger effect sizes (looking at those phenotypes with a sample size of >400k) rather than simply a greater number of causal SNPs. However, as there is a lot of dependency across the phenotypes it would be better to have these Figures made again but only including the PC-DPs. This would allow you to see if this trend exists across the most independent data points.

We have added 2 panels to Figure S5 (below). Similar to Figure 2, these plots depict effect size and p-value of the "best" SNPs for each trait plotted by either its total number of genome-wide significant loci (A) or total variance explained by the genome-wide significant loci (B), though as the reviewer suggested, restricted to independent dietary habits, PC-DPs only. The trend does persist among phenotypically independent dietary patterns.

We have also updated the text as follows:

"Across the 143 dietary traits, there was a clear positive correlation of heritability estimates (h_g^2) with the number of significant loci and the variance explained by these loci (Figure 2), a pattern that persisted when limited to phenotypically independent DP-PCs, with outliers often

explained by smaller sample size (Supplementary Figure 5).”

Referee response.

This response from the authors is good but I think it would be better for the authors to discuss what this means for the genetic architecture of these phenotypes a little more.

5. Furthermore, the large r_g between PC1 with educational attainment, and the overlap in loci identified with intelligence, indicates to me that some of what makes up choice of diet in the UK today is linked to socioeconomic status. Previous studies have shown that SES is heritable (2) as well as that intelligence is causally linked to difference in SES in UK Biobank(3). I think that the authors should discuss this link as it may indicate that PC1 is tapping both variance associated with obesity related traits, as evidenced by the 38 loci associated with obesity traits, as well as drivers of SES, as evidenced by the 24 loci associated with PC1 and with intelligence and education.

We have including some key phrases and genetic correlations to the following sections in the discussion to emphasize the overlap between PC1 and educational attainment and BMI, and between educational attainment and income and intelligence.

“While bidirectional MR demonstrates some pleiotropic effects between educational attainment and PC1 indicating either shared biology or upstream common cause(s), the relative strengths of these causal estimates suggests that higher educational attainment and/or correlated phenotypes (such as socioeconomic status [income $r_g = 0.77$] or factors related to school performance, such as fluid intelligence test scores [$r_g = 0.68$]) either directly or indirectly shift eating habits towards a healthier, more prudent diet.”

“Of note, the interpretation of the genetic component of predominantly environmental traits, such as dietary intake, can be complicated. As we have shown, dietary habits are highly correlated both with each other and with non-dietary traits, suggesting that any single dietary phenotype may represent a broader diet and lifestyle. For instance, dietary pattern PC1 with both high genetic correlation with educational attainment and large overlap in genome-wide significant loci with BMI, is capturing variance in both obesity-related traits and measures of socioeconomic status.”

We have also highlighted this in the results section when assessing overlap of top genetic signals as follows:

“Of these 83, 24 were associated with intelligence or educational attainment and not obesity-related traits; 38 were associated with obesity-related traits and not intelligence or educational attainment, indicating PC1 may be responsive to distinct sets of heritable factors related separately to educational attainment and to BMI.”

Referee response.

I am happy with the author’s response.

6. Looking at supplementary table 5 it seems that PC1 has no genetic correlation with BMI, multiple measures of adiposity, diabetes, heart disease. What do the authors think about this? Shouldn’t measures of diet be linked to disease with a large dietary component?

A closer look at the genetic correlation between PC1 and these traits revealed nominally significant, albeit small genetic correlations. Specifically PC1 is genetically correlated with BMI (field 21001_irtnt: $r_g = -0.0563$, $P=0.0243$), diabetes (self-reported any diabetes field 2443: $r_g = -0.0786$, $P=0.0013$), and ‘major coronary heart disease event’ (field I9 CHD: $r_g = -0.1809$, $P=2.27 \times 10^{-7}$). The genetic correlation matrix in Table S5 uses a strict Bonferroni-adjusted significance

threshold, reporting any non-significant genetic correlations with $P > 1.09 \times 10^{-7}$ as 0. Interestingly, a large number of SNPs that are genome-wide significant for PC1 are at least nominally significant for BMI, T2D from DIAGRAM, CAD from CARDIOGRAM
Genome-wide significant SNPs from Lookup GWAS Number of SNPs present in lookup GWAS
 $P < 0.05$ in lookup GWAS $P < 5 \times 10^{-8}$ in lookup GWAS
PC1 BMI 140 78 29
PC1 T2D 94 18 2
PC1 CAD 105 12 1
BMI PC1 1165 337 18
T2D PC1 37 3 1
CAD PC1 38 10 2

This potentially indicates that while there is a nontrivial overlap between the top SNPs affecting PC1 and BMI and cardiometabolic disease, there is a relatively low overall genetic correlation among all SNPs. This could indicate the presence of shared genetic pathways that have large effects on both PC1 and BMI and cardiometabolic disease and the presence of non-overlapping pathways specific to either PC1 or BMI and cardiometabolic disease that therefore lower the overall genetic correlation between the two respective traits.

Another key aspect to this analysis is that our dietary habits are non-isocaloric, indicating that they were not adjusted for total energy intake. As Reviewer #2 previously mentioned, adjusting for a heritable covariate leads to a series of different biases depending on the precise relationships between the genetic variants, the outcome, and the heritable covariate. However, it is important to point out that individuals who consume a large total energy intake will in turn eat more of any given food, and that may be due to a larger required energy load which is influenced by BMI and physical activity. Dietary habits that are adjusted for total energy intake might have a different, perhaps stronger genetic correlation with BMI and cardiometabolic disease. Though we indicate in the manuscript that single foods and dietary patterns display little to no correlation with total energy intake in UK Biobank, we plan to explore how BMI and physical activity affect the relationships between genetics and dietary intake in a more depth future analysis.

We appreciate the attention brought to this and have expanded on this in the follow sections of the manuscript

"In contrast to the educational attainment analyses, we were unable to provide robust evidence of a causal relationship in either direction between BMI and PC1 due to significant heterogeneous pleiotropic effects leading to inconsistent causal effect estimates (Table 2, Supplementary Figure 8). Taken together with an overlap between genome-wide significant variants and small overall genetic correlation ($r_g = -0.056$, $P = 0.024$), we postulate that BMI and our non-isocaloric PC1 have a highly complex multi-pathway relationship for which there may be overlapping genetic etiology, but a mostly-distinct underlying genetic architecture."

Referee response.

I am happy with the author's response.

Minor points

1. The authors state "The relationships between the 83 FI-QTs and the 60 PC-DPs that have significant heritability are illustrated in Supplementary Figure 4." However, no relationship between these variables is plotted, only the heritability of each. It is a valuable figure but this caption is wrong.

We agree with the reviewer and have corrected the text to say:

"Phenotypic and genetic correlation between the 60 PC-DPs and their significantly contributing FI-QTs are illustrated in Supplementary Figure 4."

Referee response.

I am happy with the author's response.

--Also why are 60 used here? It was only 20 that were selected on the basis of their eigenvalues being greater than 1.

We filtered dietary habits for downstream genetic analysis based on genetic variance explained (SNP heritability) instead of phenotypic variance explained (based on eigenvalues). While we thought it was interesting to mention the number of PCs that had eigenvalues greater than 1, we used heritability instead as it could be applied to both quantitative traits and principal components and genetic variance was more appropriate for our downstream genetic analyses.

We have clarified this in the text as follows:

“Overall, 84.1% of dietary habits analyzed (83/85 FI-QTs and 60/85 PC-DPs) were significantly heritable (as assessed by $h_g^2 P < 0.05 / 170 = 2.9 \times 10^{-4}$; see Methods, Table 1, Supplementary Table 1, and Supplementary Figure 3), displayed extensive genetic correlation (r_g ; Supplementary Table 3), and were included in downstream genomic analysis.”

Referee response.

This clarification is satisfactory.

2. Regarding the Mendelian randomisation analysis between BMI and PC1. Wouldn't you need a significant r_g between these two phenotypes?

Yes, in theory there needs to be genetic overlap between the two traits. Traditional MR analyses, as we have done here, only use the top genetic variants as the genetic instrument for the exposure. As such, we may be only capturing one aspect of PC1 biology that overlaps with top BMI genetic variants. Our MR results suggest high heterogeneity and directional pleiotropy, which could be related to the contrast seen between the strong genetic overlap between top SNPs vs low overall shared genetic architecture (i.e. genetic correlation), again consistent with the potential for complex shared and unique pathways between BMI and PC1. Future analyses using both clustered and larger sets of SNPs as genetic instruments could help tease apart these relationships, though we believe this beyond the scope of this manuscript. Please see the changes to the manuscript in Reviewer #2 comment 6 above.

Referee response.

This response is clear and I agree with the author.

Referee additional comments.

The authors have included many items in their PCA that appear to be the same for example from Figure 1. “bread type: wholemeal/wholegrain vs. any other” and “bread type: wholemeal/wholegrain vs. white + brown” which correlate 0.9986 and in Figure 4 PC3 “butter and margarine spreads vs. oil-based spreads” and “spread type: butter and butter-like spreads vs. oil-based spreads” which show a correlation of 0.998. This seems to be true for a lot of the principal components in that they contain many highly correlated items. My question is to what extent is the PCs derived biased towards capturing variance from questions that are likely to be indistinguishable to most people i.e. spread type: butter + margarine vs. never vs spread type: butter vs. never?

Furthermore, for many of these PCs (for example PC1, PC6, PC9, PC10, PC15, PC16, PC23, PC25, PC26, PC31, PC35, PC43) only a minority of items show a loading on the PCs greater than 5% indicating these are not contributing towards these components in a meaningful way. PC1 is

discussed as “Overall, the FI-QTs that have high positive loadings for PC1 include wholemeal/wholegrain bread consumption, increased fruit and vegetable intake, increased oily fish intake, and increased water intake.” But that’s not really correct. It’s really a PC that captures whether white or brown bread is eaten. I think the authors could be clearer with what their PC’s are capturing.

Reviewer #2 References

1. Aschard H, Vilhjálmsson Bjarni J, Joshi Amit D, Price Alkes L, Kraft P (2015): Adjusting for Heritable Covariates Can Bias Effect Estimates in Genome-Wide Association Studies. *The American Journal of Human Genetics*. 96:329-339.
2. Hill WD, Hagenaars SP, Marioni RE, Harris SE, Liewald DC, Davies G, et al. (2016): Molecular genetic contributions to social deprivation and household income in UK Biobank. *Current Biology*. 26:3083-3089.
3. Hill WD, Davies NM, Ritchie SJ, Skene NG, Bryois J, Bell S, et al. (2019): Genetic analysis identifies molecular systems and biological pathways associated with household income. *bioRxiv*.573691.

Referee references

1. Hill W, et al. A combined analysis of genetically correlated traits identifies 187 loci and a role for neurogenesis and myelination in intelligence. *Molecular psychiatry*, 1 (2018).
2. Davies G, et al. Study of 300,486 individuals identifies 148 independent genetic loci influencing general cognitive function. *Nature Communications* 9, 2098 (2018).
3. Savage JE, et al. Genome-wide association meta-analysis in 269,867 individuals identifies new genetic and functional links to intelligence. *Nature Genetics* 50, 912-919 (2018).
4. Hill WD, et al. Molecular genetic contributions to social deprivation and household income in UK Biobank. *Current Biology* 26, 3083-3089 (2016).
5. Hill WD, et al. Genetic analysis identifies molecular systems and biological pathways associated with household income. *bioRxiv*, 573691 (2019).

Reviewer #3:

Remarks to the Author:

The authors have addressed the comments I raised in the previous round, and I have no further comments.

NCOMMS-19-17863-A / NCOMMS-19-17863-T

We would again like to thank the reviewers for their review, and the additional constructive comments. We have included the previous response to reviewer comments below with the addition of new referee comments and author responses in blue.

Response to reviews for “Comprehensive genomic analysis of dietary habits in UK Biobank identifies hundreds of genetic loci and establishes causal relationships between educational attainment and healthy eating”

We would like to thank the reviewers for their constructive comments.

Our primary extensive change to the manuscript relates to additional analysis testing for the effects of residual confounding in our dietary habit GWAS. We have added the following analyses and text to the manuscript, and refer to this paragraph as the “primary additional analysis paragraph” throughout the review. We additionally include all areas in which we made changes to the manuscript text, *and italicize specific changes.*

“Recent work in a non-mixed model setting has demonstrated the presence of latent structure in UK Biobank co-incident with outcomes such as educational attainment, which is also associated with dietary intake (Haworth et al. and Shimakawa et al). We found that our linear mixed model dietary habit GWAS had little to no residual confounding as estimated by LD score regression intercepts (median = 1.018, inter-quartile range (IQR):1.011-1.030) and ratios (median = 0.087, IQR: 0.059-0.0123; Supplementary Table 1). Furthermore, large-scale sensitivity analysis re-conducting all 143 significantly heritable dietary habit GWAS additionally adjusting for assessment center in the genetic model only found slight attenuation of heritability (median from 4.6% to 4.2%, IQR from 2.4-7.9% to 2.2-7.4%). We report the results in this paper without additional adjustment for assessment center.”

Point-by-point responses:

Reviewer #2

- 1. For the Mendelian randomisation the fluid intelligence test score has been adjusted for the Townsend Score. As adjusting for heritable covariates can bias GWAS estimates(1) I would advise this analysis to be repeated without controlling for the Townsend Score.**

We agree that an initial analysis should avoid the adjustment of heritable covariates, which may bias results when the genetic variant is associated with the covariate (Townsend deprivation index) either alone or independently of the outcome (intelligence). We chose to initially correct for both assessment center and Townsend deprivation index in the fluid intelligence GWAS to recapitulate the genetic model used in the most recent intelligence GWAS publication (Snieder et al 2017).

The Townsend deprivation index in UK Biobank (field 189) is based on national census data and was used as an additional proxy for geographic location in addition to assessment center. As highlighted in the primary additional analysis paragraph above, we find minimal residual population stratification. This suggests to us that while assessment center and Townsend deprivation index are both significantly associated with several outcomes (including both dietary habits and fluid intelligence scores) and are co-incident with geographical genetic variation (Haworth et al 2019), linear mixed models, the gold standard GWAS analytical method, sufficiently captures residual population structure in our analyses of dietary habits in UK Biobank to avoid substantial confounding.

Nevertheless, we agree with the value in performing the suggested sensitivity analysis and demonstrate below highly concordant results with and without adjustment for Townsend deprivation and assessment center on fluid intelligence test score heritability, GWAS, and Mendelian randomization with PC1.

	Original INT results (adjustment)	Sensitivity analysis INT results (no adjustment)
SNP Heritability (h2g)	26.7%	26.9%
Genome-wide significant SNPs	184	175
Weighted Median Mendelian Randomization Exposure: INT Outcome: PC1	Estimate = 0.261 P-value = 7.71×10^{-13}	Estimate = 0.278 P-value = 1.15×10^{-13}
Weighted Median Mendelian Randomization Exposure: PC1 Outcome: INT	Estimate = 0.074 P-value = 8.04×10^{-7}	Estimate = 0.110 P-value = 1.10×10^{-12}

Referee response:

The Sniekers et al paper is not the most recent GWAS of intelligence as this only includes data from the initial release of the UK Biobank genetic data. Since then the full UK Biobank cognitive data has been published as part of a large GWAS¹ and this has been followed by replication GWASs.^{2,3}

Furthermore, the authors state that the Townsend score is an additional proxy for geographic location in a similar manner to assessment centre. Whilst the authors are correct that the Townsend score captures geographical genetic variation, this is not due to it capturing population stratification issues, but rather indicates that the heritable traits linked to differences in socioeconomic status are not distributed evenly across the UK.^{4, 5} However, the sensitivity analysis conducted by the author's shows that these covariates make very little difference to the analyses. It would be better however, if they could only omit the Townsend score rather than the Townsend score and the assessment centre.

Author response:

Thank you for bringing those references to our attention. We have now repeated the above analysis with the inclusion of assessment center and the exclusion of townsend deprivation index with highly similar genetic results. The newest intelligence GWAS has similar SNP heritability and number of genome-wide significance SNPs, and as such we do not feel a need to repeat the MR analysis as well (see final far right column in table below).

	Original INT results (adjustment for both townsend deprivation + assessment center)	Sensitivity analysis INT results (no adjustment for either townsend deprivation or assessment center)	New INT results (adjusting for assessment center and not townsend deprivation)
SNP Heritability (h2g)	26.7%	26.9%	26.2%
Genome-wide significant SNPs	184	175	186

2. Including 10 PCs in the LMM seems low.

We believe that including PCs on top of a linear mixed model is fairly uncommon and rather conservative. However, we chose to include 10 PCs that were re-derived from our European only sample (as opposed to using the multi-ancestry PCs provided by UKB) as an added precautionary measure and to speed up the mixed model computation. As depicted below in our PC bi-plots, our lower order PCs capture little to no additional variation.

We also include below a fish plot comparing significance and directionality of dietary pattern PC1 LMM GWAS results with 10 PCs vs. 40 PCs. P-values are signed based on consistency in the direction of the betas between the two comparison models. Thus quadrant 1 represents $-\log(P\text{-values})$ of SNPs with effect sizes in the same direction and quadrant 3 represents $-\log(P\text{-values})$ of SNPs with effect sizes in opposite directions. The vast majority of SNPs fall very close to the line of identity in quadrant 1, and thus there is minimal residual confounding not captured by the linear mixed model and 10 PCs.

FISH Plot - PC1 - BOLTImm with 10PCs vs 40PCs

-- The LDSC regression intercepts should be made available too to ascertain if there is any residual stratification.

LD score regression (LDSC) intercepts and ratios are now included in Table S1 and the inclusion of this analysis is described in the primary additional analysis paragraph above. The 6-number descriptive summary of the LDSC intercepts and ratios for our heritable traits are summarized below, essentially demonstrating little to no residual confounding that is not controlled for with our linear mixed model as the intercepts are near one and the bulk of the ratios are near zero (Bulik-Sullivan et al 2015). Only PC62 had a ratio greater than 40%, and only five dietary habits had ratios greater than 30% indicating mild residual confounding, including PC40, PC44, PC48, PC62, and never eat wheat vs. no wheat restrictions.

	Min	Q1	Median	Mean	Q3	Max
LDSC intercept	0.993	1.011	1.018	1.021	1.030	1.074
LDSC ratio	0.00	0.059	0.087	0.102	0.123	0.599

Referee response.

I am very happy with this thorough response.

3. Too often statements are made that compare two estimates of heritability but no formal test is ever used to test if this difference is significant.

We have added formal tests comparing heritability estimates between PC1 and its 19 contributing QTs and updated the text as follows:

“The first PC-DP (hereafter referred to as PC1) is among the most heritable dietary patterns (PC1 $h_g^2=13.6\%$, Table 1), and is more heritable than all of its individual contributing FI-QTs (*all h_g^2 comparisons have $P < 5.0 \times 10^{-45}$*).”

We have added the following sentence to the Methods

“Z-score calculations were used to test for significant differences in heritability between PC1 and its 19 contributing QTs ($Z = \frac{(h_{g-PC1}^2 - h_{g-QT}^2)}{\sqrt{SE-PC1^2 + SE-QT^2}}$)”

Referee response.

The authors have addressed my concerns here.

4. The caption for Supplementary Figure 5A and Supplementary Figure 5B has been swapped.

We have updated Figure 5 to correctly align with the caption and appreciate the reviewer bringing this to our attention.

--The finding presented here is, I think, quite interesting as it is saying that more heritable phenotypes have SNPs with larger effect sizes (looking at those phenotypes with a sample size of >400k) rather than simply a greater number of causal SNPs. However, as there is a lot of dependency across the phenotypes it would be better to have these Figures made again but only including the PC-DPs. This would allow you to see if this trend exists across the most independent data points.

We have added 2 panels to Figure S5 (below). Similar to Figure 2, these plots depict effect size and p-value of the “best” SNPs for each trait plotted by either its total number of genome-wide significant loci (A) or total variance explained by the genome-wide significant loci (B), though as the reviewer suggested, restricted to independent dietary habits, PC-DPs only. The trend does persist among phenotypically independent dietary patterns.

We have also updated the text as follows:

“Across the 143 dietary traits, there was a clear positive correlation of heritability estimates (h_g^2) with the number of significant loci and the variance explained by these loci (Figure 2), a pattern that persisted when limited to phenotypically independent DP-PCs, with outliers often explained by smaller sample size (Supplementary Figure 5).”

Referee response.

This response from the authors is good but I think it would be better for the authors to discuss what this means for the genetic architecture of these phenotypes a little more.

Author response:

We agree with your interpretation and appreciate your attention in bringing it to our manuscript. We have added the following sentence to the manuscript

“We also find evidence that the more heritable phenotypes have lead SNPs with larger effect sizes rather than simply a greater number of significant loci (Figure 2 and Supplementary Figure 5).”

5. Furthermore, the large rg between PC1 with educational attainment, and the overlap in loci identified with intelligence, indicates to me that some of what makes up choice of diet in the UK today is linked to socioeconomic status. Previous studies have shown that SES is heritable (2) as well as that intelligence is causally linked to difference in SES in UK Biobank(3). I think that the authors should discuss this link as it may indicate that PC1 is tapping both variance associated with obesity related traits, as evidenced by the 38 loci associated with obesity traits, as well as drivers of SES, as evidenced by the 24 loci associated with PC1 and with intelligence and education.

We have including some key phrases and genetic correlations to the following sections in the discussion to emphasize the overlap between PC1 and educational attainment and BMI, and between educational attainment and income and intelligence.

“While bidirectional MR demonstrates some pleiotropic effects between educational attainment and PC1 *indicating either shared biology or upstream common cause(s)*, the relative strengths of these causal estimates suggests that higher educational attainment and/or correlated phenotypes (such as socioeconomic status [*income* $r_g = 0.77$] or factors related to school performance, *such as fluid intelligence test scores* [$r_g = 0.68$]) *either directly or indirectly* shift eating habits towards a healthier, more prudent diet.”

“Of note, the interpretation of the genetic component of predominantly environmental traits, such as dietary intake, can be complicated. As we have shown, dietary habits are highly correlated both with each other and with non-dietary traits, suggesting that any single dietary phenotype may represent a broader diet and lifestyle. For instance, dietary pattern PC1 with both high genetic correlation with educational attainment and large overlap in genome-wide significant loci with BMI, is capturing variance in both obesity-related traits and measures of socioeconomic status.”

We have also highlighted this in the results section when assessing overlap of top genetic signals as follows:

“Of these 83, 24 were associated with intelligence or educational attainment and not obesity-related traits; 38 were associated with obesity-related traits and not intelligence or educational attainment, *indicating PC1 may be responsive to distinct sets of heritable factors related separately to educational attainment and to BMI.*”

Referee response.

I am happy with the author’s response.

6. Looking at supplementary table 5 it seems that PC1 has no genetic correlation with BMI, multiple measures of adiposity, diabetes, heart disease. What do the authors think about this? Shouldn’t measures of diet be linked to disease with a large dietary component?

A closer look at the genetic correlation between PC1 and these traits revealed nominally significant, albeit small genetic correlations. Specifically PC1 is genetically correlated with BMI (field 21001_irnt: $r_g = -0.0563$, $P = 0.0243$), diabetes (self-reported any diabetes field 2443: $r_g = -0.0786$, $P = 0.0013$), and ‘major coronary heart disease event’ (field I9 CHD: $r_g = -0.1809$, $P = 2.27 \times 10^{-7}$). The genetic correlation matrix in Table S5 uses a strict Bonferroni-adjusted significance threshold, reporting any non-significant genetic correlations with $P > 1.09 \times 10^{-7}$ as 0.

Interestingly, a large number of SNPs that are genome-wide significant for PC1 are at least nominally significant for BMI, T2D from DIAGRAM, CAD from CARDIOGRAM

Genome-wide significant SNPs	Lookup GWAS	Number of SNPs present in lookup	$P < 0.05$ in lookup	$P < 5 \times 10^{-8}$ in
-------------	----------------------------------	----------------------	---------------------------

from		GWAS	GWAS	lookup GWAS
PC1	BMI	140	78	29
PC1	T2D	94	18	2
PC1	CAD	105	12	1
BMI	PC1	1165	337	18
T2D	PC1	37	3	1
CAD	PC1	38	10	2

This potentially indicates that while there is a nontrivial overlap between the top SNPs affecting PC1 and BMI and cardiometabolic disease, there is a relatively low overall genetic correlation among all SNPs. This could indicate the presence of shared genetic pathways that have large effects on both PC1 and BMI and cardiometabolic disease and the presence of non-overlapping pathways specific to either PC1 or BMI and cardiometabolic disease that therefore lower the overall genetic correlation between the two respective traits.

Another key aspect to this analysis is that our dietary habits are non-isocaloric, indicating that they were not adjusted for total energy intake. As Reviewer #2 previously mentioned, adjusting for a heritable covariate leads to a series of different biases depending on the precise relationships between the genetic variants, the outcome, and the heritable covariate. However, it is important to point out that individuals who consume a large total energy intake will in turn eat more of any given food, and that may be due to a larger required energy load which is influenced by BMI and physical activity. Dietary habits that are adjusted for total energy intake might have a different, perhaps stronger genetic correlation with BMI and cardiometabolic disease. Though we indicate in the manuscript that single foods and dietary patterns display little to no correlation with total energy intake in UK Biobank, we plan to explore how BMI and physical activity affect the relationships between genetics and dietary intake in a more depth future analysis.

We appreciate the attention brought to this and have expanded on this in the follow sections of the manuscript

“In contrast to the educational attainment analyses, we were unable to provide robust evidence of a causal relationship in either direction between BMI and PC1 due to significant heterogeneous pleiotropic effects leading to inconsistent causal effect estimates (Table 2, Supplementary Figure 8). *Taken together with an overlap between genome-wide significant variants and small overall genetic correlation ($r_g = -0.056$, $P = 0.024$), we postulate that BMI and our non-isocaloric PC1 have a highly complex multi-pathway relationship for which there may be overlapping genetic etiology, but a mostly-distinct underlying genetic architecture.*”

Referee response.

I am happy with the author’s response.

Minor points

1. The authors state “The relationships between the 83 FI-QTs and the 60 PC-DPs that have significant heritability are illustrated in Supplementary Figure 4.” However, no

relationship between these variables is plotted, only the heritability of each. It is a valuable figure but this caption is wrong.

We agree with the reviewer and have corrected the text to say:

“Phenotypic and genetic correlation between the 60 PC-DPs and their significantly contributing FI-QTs are illustrated in Supplementary Figure 4.”

Referee response.

I am happy with the author’s response.

--Also why are 60 used here? It was only 20 that were selected on the basis of their eigenvalues being greater than 1.

We filtered dietary habits for downstream genetic analysis based on genetic variance explained (SNP heritability) instead of phenotypic variance explained (based on eigenvalues). While we thought it was interesting to mention the number of PCs that had eigenvalues greater than 1, we used heritability instead as it could be applied to both quantitative traits and principal components and genetic variance was more appropriate for our downstream genetic analyses.

We have clarified this in the text as follows:

“Overall, 84.1% of dietary habits analyzed (83/85 FI-QTs and 60/85 PC-DPs) were significantly heritable (as assessed by h_g^2 $P < 0.05 / 170 = 2.9 \times 10^{-4}$; see Methods, Table 1, Supplementary Table 1, and Supplementary Figure 3), displayed extensive genetic correlation (r_g ; Supplementary Table 3), and were included in downstream genomic analysis.”

Referee response.

This clarification is satisfactory.

2. Regarding the Mendelian randomisation analysis between BMI and PC1. Wouldn’t you need a significant r_g between these two phenotypes?

Yes, in theory there needs to be genetic overlap between the two traits. Traditional MR analyses, as we have done here, only use the top genetic variants as the genetic instrument for the exposure. As such, we may be only capturing one aspect of PC1 biology that overlaps with top BMI genetic variants. Our MR results suggest high heterogeneity and directional pleiotropy, which could be related to the contrast seen between the strong genetic overlap between top SNPs vs low overall shared genetic architecture (i.e. genetic correlation), again consistent with the potential for complex shared and unique pathways between BMI and PC1. Future analyses using both clustered and larger sets of SNPs as genetic instruments could help tease apart these relationships, though we believe this beyond the scope of this manuscript. Please see the changes to the manuscript in Reviewer #2 comment 6 above.

Referee response.

This response is clear and I agree with the author.

Referee additional comments.

1. The authors have included many items in their PCA that appear to be the same for example from Figure 1. “bread type: wholemeal/wholegrain vs. any other” and “bread type: wholemeal/wholegrain vs. white + brown” which correlate 0.9986 and in Figure 4 PC3 “butter and margarine spreads vs. oil-based spreads” and “spread type: butter and butter-like spreads vs. oil-based spreads” which show a correlation of 0.998. This seems to be true for a lot of the principal components in that they contain many highly correlated items. My question is to what extent is the PCs derived biased towards capturing variance from questions that are likely to be indistinguishable to most people i.e. spread type: butter + margarine vs. never vs spread type: butter vs. never?

Author response

We agree there is an extensive amount of phenotypic correlation between FI-QTs that were included in the PCA. While the addition of correlated variables will affect the total percentage of variance explained by each PC, it will not affect the resulting PCs and their respective combinations of foods. Therefore, we chose to take a more data-driven inclusive approach, and retain all quantitative traits in the PCA. This is yet another good reason to use heritability of each trait as a filter as opposed to a percentage of variance explained cutoff. We also hope including phenotypic correlation in Figure 1 and Supplementary Figure 4 allows the reader to view these correlated relationships within each PC. We have added the following text to the manuscript to explain this:

“While the inclusion of highly correlated FI-QTs in the PCA will affect the total percentage of variance explained by each PC, it will not affect the make-up of foods contributing to each PC. Therefore, we used heritability of all 85 FI-QTs and 85 PC-DPs as a filter for downstream genomic analysis.”

2. Furthermore, for many of these PCs (for example PC1, PC6, PC9, PC10, PC15, PC16, PC23, PC25, PC26, PC31, PC35, PC43) only a minority of items show a loading on the PCs greater than 5% indicating these are not contributing towards these components in a meaningful way. PC1 is discussed as “Overall, the FI-QTs that have high positive loadings for PC1 include wholemeal/wholegrain bread consumption, increased fruit and vegetable intake, increased oily fish intake, and increased water intake.” But that’s not really correct. It’s really a PC that captures whether white or brown bread is eaten. I think the authors could be clearer with what their PC’s are capturing.

Author response

It is the case that some of the PCs are largely driven by a small number of FI-QTs, and that including highly correlated FI-QTs will decrease their individual percent contribution to the PC.

For instance, four highly correlated and overlapping bread-related FI-QTs each contribute 10-15% to PC1, together totally at most 60% contribution. The other 40% of PC1 is largely made up of the other FI-QTs that pass our significance threshold for inclusion in Figure 1. We use a simple cutoff for this in Figure 1 and S4 as described in the Methods section. (“FI-QTs with percent contribution (squared coordinates) greater than expected under a uniform distribution [$1/85*100=1.18\%$] were included in Figure 1 and Supplementary Figure 4”).

We believe we do in fact put appropriate emphasis on the contribution of type of bread on PC1 throughout the text as follows:

- “PC1 ... is primarily defined by the type of bread consumed (wholegrain/wholemeal vs. white bread). Overall, the FI-QTs that have high positive loadings for PC1 include wholemeal/wholegrain bread consumption, increased fruit and vegetable intake, increased oily fish intake, and increased water intake. The FI-QTs that have high negative loadings include white bread consumption, butter and oil spread consumption, increased processed meat intake, and consumption of milk with higher fat content (Figure 1).”
- “Finally, the aspects of a prudent dietary pattern reflected by PC1 (predominantly driven by wholemeal/wholegrain vs. white bread consumption) may not capture the causal protective features of a prudent dietary pattern.”

Furthermore, we include phenotypic correlation of the FI-QTs for each PC in Figure 1 and Supplementary Figure 4 to demonstrate the overlap among some of the FI-QTs.

Interestingly, when we repeat the MR analysis of educational attainment with the 19 FI-QTs that significantly contribute to PC1 and compare the results to educational attainment and PC1, educational attainment has the largest effect on PC1 (with non-overlapping CI with the 19 FI-QTs), indicating true effects on PC1 beyond just bread consumption.

Weighted Median Results with Educational Attainment Exposure

Outcome	Estimate	CI lower	CI upper	P-value
PC1	0.823	0.751	0.894	6.59E-113
bread type: wholemeal/wholegrain vs. white + brown	0.397	0.362	0.433	1.97E-106
bread type: wholemeal/wholegrain vs. any other	0.366	0.332	0.401	2.59E-95
bread type: white vs. any other	-0.357	-0.389	-0.325	6.18E-107
bread type: white vs. wholemeal/wholegrain + brown	-0.363	-0.396	-0.330	1.10E-101
pieces of fresh fruit per day	0.081	0.050	0.113	4.18E-07
spread type: all spreads vs. never	-0.050	-0.071	-0.028	7.65E-06
spread type: butter + margarine vs. never	-0.090	-0.122	-0.057	4.91E-08
spread type: butter vs. never	-0.077	-0.118	-0.036	0.000232102
overall processed meat intake	-0.017	-0.047	0.012	0.252382614

milk type: skimmed, semi-skimmed, full cream (QT)	0.049	0.021	0.078	0.000685274
overall oily fish intake	0.141	0.111	0.172	9.30E-20
tablespoons of raw vegetables per day	0.015	-0.015	0.045	0.342731568
pieces of dried fruit per day	0.253	0.223	0.283	4.61E-62
spread type: any oil based spread vs. never	-0.066	-0.111	-0.021	0.004317514
spread type: butter vs. any other	0.032	-0.001	0.066	0.061011324
spread type: other oil-based spread vs. never	-0.167	-0.225	-0.109	1.86E-08
spread type: olive oil spread vs. never	0.015	-0.051	0.081	0.655263859
glasses of water per day	0.072	0.041	0.102	5.18E-06
tablespoons of cooked vegetables per day	0.047	0.017	0.076	0.002009479

Reviewer #2 References

1. Aschard H, Vilhjálmsón Bjarni J, Joshi Amit D, Price Alkes L, Kraft P (2015): Adjusting for Heritable Covariates Can Bias Effect Estimates in Genome-Wide Association Studies. *The American Journal of Human Genetics*. 96:329-339.
2. Hill WD, Hagenaars SP, Marioni RE, Harris SE, Liewald DC, Davies G, et al. (2016): Molecular genetic contributions to social deprivation and household income in UK Biobank. *Current Biology*. 26:3083-3089.
3. Hill WD, Davies NM, Ritchie SJ, Skene NG, Bryois J, Bell S, et al. (2019): Genetic analysis identifies molecular systems and biological pathways associated with household income. [bioRxiv.573691](https://doi.org/10.1101/573691).

Referee additional references

1. Hill W, et al. A combined analysis of genetically correlated traits identifies 187 loci and a role for neurogenesis and myelination in intelligence. *Molecular psychiatry*, 1 (2018).
2. Davies G, et al. Study of 300,486 individuals identifies 148 independent genetic loci influencing general cognitive function. *Nature Communications* 9, 2098 (2018).
3. Savage JE, et al. Genome-wide association meta-analysis in 269,867 individuals identifies new genetic and functional links to intelligence. *Nature Genetics* 50, 912-919 (2018).
4. Hill WD, et al. Molecular genetic contributions to social deprivation and household income in UK Biobank. *Current Biology* 26, 3083-3089 (2016).
5. Hill WD, et al. Genetic analysis identifies molecular systems and biological pathways associated with household income. [bioRxiv](https://doi.org/10.1101/573691), 573691 (2019).

Reviewer #3

Reviewer #3 (Remarks to the Author): The authors have addressed the comments I raised in the previous round, and I have no further comments.

Reviewer #3 original reviews below. No additional changes have been made.

1. FFQ data are known to be noisy / unreliable measures of actual dietary intake, although they are still widely used given their ease of administration. However, this means that interpreting the results of a GWAS is complicated because the questionnaires may be picking up other phenotypes (e.g., memory, the tendency to self-report in a favourable way, etc.) than the intended phenotype. For this reason, interpreting the results of the GWAS results is challenging, and assuming that the GWAS-significant hits necessarily reflect underlying biology (see Gage et al., 2016) may be problematic.

We wholeheartedly agree that interpretation of GWAS findings with lifestyle factors is complex. As the reviewer mentions, questionnaire responses may capture memory or social desirability, while dietary intake itself is correlated with what may be hundreds of other traits, including physical activity, BMI, socioeconomic status, disease status, geographical location, climate, employment, etc. Even “simple” case-control disease phenotypes may suffer from biases relating to self-reported diagnosis, lifestyle intervention, and pre-clinical disease status. Environmental traits are nearly impossible to capture with certainty, and thus we must rely on studies that use surrogate best-available measures while presenting these limitations. We appreciate this point and have emphasized this throughout the revision to our manuscript as follows:

- 1) *“Of note, the interpretation of the genetic component of predominantly environmental traits, such as dietary intake, can be complicated. As we have shown, dietary habits are highly correlated both with each other and with non-dietary traits, suggesting that any single dietary phenotype may represent a broader diet and lifestyle. For instance, dietary pattern PC1 with both high genetic correlation with educational attainment and large overlap in genome-wide significant loci with BMI, is capturing variance in both obesity-related traits and measures of socioeconomic status. Additionally, similar to most nutritional epidemiology studies, measures of dietary intake in UKB are based on self-reported questionnaire data, which intrinsically suffers biases based on memory and favorable reporting (Smith 1991, Hebert 1995), further complicating the interpretation of genomic results.”*
- 2) *“The large and broad overlap in both the significant loci and the overall genetic make-up of our dietary habits with other traits, many of which do not have well-established biological links with diet, emphasizes a need for exploring correlation vs. causation. ... Therefore, we sought to understand the cause-and-effect relationships and degree of pleiotropy between PC1 and educational attainment, fluid intelligence scores, and BMI using bidirectional MR.” (Davey Smith & Hemani 2014)*
- 3) *“Importantly, because the instrumental variables used for educational attainment are not mechanistically linked directly to educational attainment/intelligence, it remains possible that causal influences on PC1 could be due to unmeasured heritable factor(s) that are themselves causal for educational attainment/intelligence. Furthermore, because PC1 is highly correlated with hundreds of additional outcomes, educational attainment (or traits correlated with educational attainment) could be influencing a larger complex phenotype made of many lifestyle factors that is captured here by PC1 dietary preferences.”*

- 4) “To test whether the “prudent” PC1 dietary pattern is *capturing a lifestyle that is likely to causally influence disease risk....*”
- 5) “While bidirectional MR demonstrates some pleiotropic effects between educational attainment and PC1 *indicating either shared biology or upstream common cause(s)*, the relative strengths of these causal estimates suggests that higher educational attainment and/or correlated phenotypes (such as socioeconomic status [$r_g = 0.77$] or factors related to school performance, *such as fluid intelligence test scores [$r_g = 0.68$]*) *either directly or indirectly shift eating habits towards a healthier, more prudent diet.*”

--This issue is compounded by the fact that there is latent structure within the UK Biobank genetic data that reflects geographical structure (and by extension socioeconomic confounding) (Haworth et al., 2019). This can induce bias in observed associations, even in Mendelian randomization analyses. The fact that one of the main drivers of the primary principal component of interest – type of bread consumed – is so strong socially patterned within the UK raises the possibility that the genetic variants identified are also capturing socioeconomic confounding (I strongly suspect this phenotype will be patterned by region).

We believe that our analysis using linear mixed models minimizes the effects of residual population structure in UK Biobank not captured using traditional simple regression approaches with genetic PCs alone. We have formally tested this by including LDSC intercepts and ratios which demonstrate little to no residual confounding (see the primary additional analysis paragraph and response to Reviewer #2 comment #2 above). We have additionally conducted a sensitivity analysis on all 143 significantly heritable dietary habits by including assessment center in the linear mixed model as a covariate and found only a slight attenuation of genomic signal (see primary additional analysis paragraph above and PC1-specific results in table below):

	Original PC1 results (no center adjustment)	Sensitivity analysis PC1 results (center adjustment)
SNP Heritability (h²_g)	13.5%	13.0%
Genome-wide significant SNPs	140	126
Weighted Median Mendelian Randomization Exposure: SES Outcome: PC1	Estimate = 0.823 P-value = 6.59×10^{-113}	Estimate = 0.807 P-value = 2.30×10^{-110}
Weighted Median Mendelian Randomization Exposure: PC1 Outcome: SES	Estimate = 0.199 P-value = 7.44×10^{-56}	Estimate = 0.194 P-value = 1.98×10^{-49}

We added the following text and results to the manuscript:

“Results were also in high agreement with a repeated MR analysis using our sensitivity analysis PC1 GWAS results adjusted for assessment center (Supplementary Table 7).”

Furthermore, we present our results between PC1 and educational attainment in light of significant bidirectional MR effects, indicating that PC1 is in fact capturing some of the same biology that pertains to traits correlated with educational attainment. We have revised our interpretation of results to clarify this key point as laid out in previous responses to reviewer comments above.

- 2. The paper might be strong focused on developing a tool that captures variation in dietary behaviour (although with the caveats associated with my first point) and focusing on using this to understand causal pathways to and from dietary behaviour. The relative lack of clear evidence of a causal pathway from diet to relevant health outcomes is a cause for concern however, and reinforces the point that the data may be capturing other phenotypes than simply dietary behaviour. Other approaches are possible within UK Biobank (e.g. Davies et al., 2018), but this would be quite a different study.**

We agree the one of the most likely causes for a lack of clear evidence from PC1 to cardiometabolic disease is related to the complex lifestyle that PC1 is capturing as a dietary habit. It is predominantly driven by bread type and is highly phenotypically and genetically correlated with hundreds of other outcomes. We have updated the text extensively to clarify this complication to interpretation (highlighted throughout the above reviewer comments), including an additional paragraph in the Discussion section (see reviewer #3, comment #1, update #1). We believe the development of a tool to test the causal relationships between variation in dietary intake and health outcomes is a valuable next step and beyond the scope of this manuscript. We also agree, as indicated by Davies et al, that the years of schooling educational attainment measurement is indirectly capturing other correlated measures that may confound the true effects of education on health outcomes. Our manuscript has been revised to reflect these limitations to interpretation. Those changes are highlighted above throughout this response to reviewer comments. As suggested, an analysis using the educational reform changes in the UK unaffected by genomic confounding (Davies et al 2018), is both an intriguing approach and beyond the scope of this genomic analysis.

Reviewer #3 References

Davies et al. (2018). The causal effects of education on health outcomes in the UK Biobank. *Nature Human Behaviour*, 2, 117-125.

Gage et al. (2016) G=E: What GWAS can tell us about the environment. *PLOS Genetics*, 12, e1005765.

Haworth et al. (2019). Apparent latent structure within the UK Biobank sample has implications for epidemiological analysis. *Nature Communications*, 10, 333.

Response to Reviewer Comments References

Bulik-Sullivan, B. K., Loh, P. R., Finucane, H. K., Ripke, S., Yang, J., Patterson, N., ... & Schizophrenia Working Group of the Psychiatric Genomics Consortium. (2015). LD Score

regression distinguishes confounding from polygenicity in genome-wide association studies. *Nature genetics*, 47(3), 291.

Davey Smith, G., & Hemani, G. (2014). Mendelian randomization: genetic anchors for causal inference in epidemiological studies. *Human molecular genetics*, 23(R1), R89-R98.

Davies et al. (2018). The causal effects of education on health outcomes in the UK Biobank. *Nature Human Behaviour*, 2, 117-125.

Haworth, S., Mitchell, R., Corbin, L., Wade, K. H., Dudding, T., Budu-Aggrey, A., ... & Davies, N. (2019). Apparent latent structure within the UK Biobank sample has implications for epidemiological analysis. *Nature communications*, 10(1), 333.

Hebert, J. R., Clemow, L., Pbert, L., Ockene, I. S., & Ockene, J. K. (1995). Social desirability bias in dietary self-report may compromise the validity of dietary intake measures. *International journal of epidemiology*, 24(2), 389-398.

Shimakawa, T., Sorlie, P., Carpenter, M. A., Dennis, B., Tell, G. S., Watson, R., & Williams, O. D. (1994). Dietary intake patterns and sociodemographic factors in the Atherosclerosis Risk in Communities Study. *Preventive medicine*, 23(6), 769-780.

Smith, A. F., Jobe, J. B., & Mingay, D. J. (1991).. *Applied Cognitive Psychology*, 5(3), 269-296.

Sniekers, S., Stringer, S., Watanabe, K., Jansen, P. R., Coleman, J. R., Krapohl, E., ... & Amin, N. (2017). Genome-wide association meta-analysis of 78,308 individuals identifies new loci and genes influencing human intelligence. *Nature genetics*, 49(7), 1107.

Referee additional comments.

1. The authors have included many items in their PCA that appear to be the same for example from Figure 1. “bread type: wholemeal/wholegrain vs. any other” and “bread type: wholemeal/wholegrain vs. white + brown” which correlate 0.9986 and in Figure 4 PC3 “butter and margarine spreads vs. oil-based spreads” and “spread type: butter and butter-like spreads vs. oil-based spreads” which show a correlation of 0.998. This seems to be true for a lot of the principal components in that they contain many highly correlated items. My question is to what extent is the PCs derived biased towards capturing variance from questions that are likely to be indistinguishable to most people i.e. spread type: butter + margarine vs. never vs spread type: butter vs. never?

Author response

We agree there is an extensive amount of phenotypic correlation between FI-QTs that were included in the PCA. While the addition of correlated variables will affect the total percentage of variance explained by each PC, it will not affect the resulting PCs and their respective combinations of foods. Therefore, we chose to take a more data-driven inclusive approach, and retain all quantitative traits in the PCA. This is yet another good reason to use heritability of each trait as a filter as opposed to a percentage of variance explained cutoff. We also hope including phenotypic correlation in Figure 1 and Supplementary Figure 4 allows the reader to view these correlated relationships within each PC. We have added the following text to the manuscript to explain this:

“While the inclusion of highly correlated FI-QTs in the PCA will affect the total percentage of variance explained by each PC, it will not affect the make-up of foods contributing to each PC. Therefore, we used heritability of all 85 FI-QTs and 85 PC-DPs as a filter for downstream genomic analysis.”

2. Furthermore, for many of these PCs (for example PC1, PC6, PC9, PC10, PC15, PC16, PC23, PC25, PC26, PC31, PC35, PC43) only a minority of items show a loading on the PCs greater than 5% indicating these are not contributing towards these components in a meaningful way. PC1 is discussed as “Overall, the FI-QTs that have high positive loadings for PC1 include wholemeal/wholegrain bread consumption, increased fruit and vegetable intake, increased oily fish intake, and increased water intake.” But that’s not really correct. It’s really a PC that captures whether white or brown bread is eaten. I think the authors could be clearer with what their PC’s are capturing.

Author response

It is the case that some of the PCs are largely driven by a small number of FI-QTs, and that including highly correlated FI-QTs will decrease their individual percent contribution to the PC. For instance, four highly correlated and overlapping bread-related FI-QTs each contribute 10-15% to PC1, together totally at most 60% contribution. The other 40% of PC1 is largely made up of the other FI-QTs that pass our significance threshold for inclusion in Figure 1. We use a simple cutoff for this in Figure 1 and S4 as described in the Methods section. (“FI-QTs with percent contribution (squared coordinates) greater than expected under a uniform distribution [$1/85*100=1.18\%$] were included in Figure 1 and Supplementary Figure 4”).

We believe we do in fact put appropriate emphasis on the contribution of type of bread on PC1 throughout the text as follows:

- “PC1 ... is primarily defined by the type of bread consumed (wholegrain/wholemeal vs. white bread). Overall, the FI-QTs that have high positive loadings for PC1 include wholemeal/wholegrain bread consumption, increased fruit and vegetable intake, increased oily fish intake, and increased water intake. The FI-QTs that have high negative loadings include white bread consumption, butter and oil spread consumption, increased processed meat intake, and consumption of milk with higher fat content (Figure 1).”
- “Finally, the aspects of a prudent dietary pattern reflected by PC1 (predominantly driven by wholemeal/wholegrain vs. white bread consumption) may not capture the causal protective features of a prudent dietary pattern.”

Furthermore, we include phenotypic correlation of the FI-QTs for each PC in Figure 1 and Supplementary Figure 4 to demonstrate the overlap among some of the FI-QTs.

Interestingly, when we repeat the MR analysis of educational attainment with the 19 FI-QTs that significantly contribute to PC1 and compare the results to educational attainment and PC1, educational attainment has the largest effect on PC1 (with non-overlapping CI with the 19 FI-QTs), indicating true effects on PC1 beyond just bread consumption.

Weighted Median Results with Educational Attainment Exposure

Outcome	Estimate	CI lower	CI upper	P-value
PC1	0.823	0.751	0.894	6.59E-113
bread type: wholemeal/wholegrain vs. white + brown	0.397	0.362	0.433	1.97E-106
bread type: wholemeal/wholegrain vs. any other	0.366	0.332	0.401	2.59E-95
bread type: white vs. any other	-0.357	-0.389	-0.325	6.18E-107
bread type: white vs. wholemeal/wholegrain + brown	-0.363	-0.396	-0.330	1.10E-101
pieces of fresh fruit per day	0.081	0.050	0.113	4.18E-07
spread type: all spreads vs. never	-0.050	-0.071	-0.028	7.65E-06
spread type: butter + margarine vs. never	-0.090	-0.122	-0.057	4.91E-08
spread type: butter vs. never	-0.077	-0.118	-0.036	0.000232102
overall processed meat intake	-0.017	-0.047	0.012	0.252382614
milk type: skimmed, semi-skimmed, full cream (QT)	0.049	0.021	0.078	0.000685274
overall oily fish intake	0.141	0.111	0.172	9.30E-20
tablespoons of raw vegetables per day	0.015	-0.015	0.045	0.342731568
pieces of dried fruit per day	0.253	0.223	0.283	4.61E-62
spread type: any oil based spread vs. never	-0.066	-0.111	-0.021	0.004317514
spread type: butter vs. any other	0.032	-0.001	0.066	0.061011324
spread type: other oil-based spread vs. never	-0.167	-0.225	-0.109	1.86E-08
spread type: olive oil spread vs. never	0.015	-0.051	0.081	0.655263859
glasses of water per day	0.072	0.041	0.102	5.18E-06
tablespoons of cooked vegetables per day	0.047	0.017	0.076	0.002009479

Reviewer response to both of the comments above.

The authors write “*While the addition of correlated variables will affect the total percentage of variance explained by each PC*” this is the issue. The PC analysis attempts to explain a large number of variables with a smaller number of components based on how the variables correlate. By including multiple measures of the same item (e.g. bread) the components derived by the authors will be largely composed of these repeated items. The first component only explains 8.63% of the variance and this is almost entirely driven by four questions about bread. However, this component is often called western Vs Prudent diet which is a little misleading looking at how much bread contributes to this factor. Another effect of including multiple items that are measuring the same thing is that the resulting component will be biased towards this subset of highly correlated items and lower the loadings on other items. For example, if a principal component analysis was conducted on five items measuring bread consumption and one of fruit consumption, the resulting component would have an axis closer to the cluster of bread items and further from the fruit. Repeat the analysis with one bread item and one fruit item and the axis will be closer to the fruit item. This shows that what the authors believe when they state “it will not affect the resulting PCs” is not actually correct.

I also disagree when the authors state that “*Overall, the FI-QTs that have high positive loadings for PC1 include wholemeal/wholegrain bread consumption, increased fruit and vegetable intake, increased oily fish intake, and increased water intake. The FI-QTs that have high negative loadings include white bread consumption, butter and oil spread consumption, increased processed meat intake, and consumption of milk with higher fat content (Figure 1).*” Looking at Figure 1. Bread items contribute between 10 and 15% but each of the non-bread items contributes less than 2.5%, which is in no way a high contribution. Again describing this as a Western vs. prudent diet is inaccurate as, it’s just bread.

The authors seem very keen to describe their results in terms of identifying these components but I don’t believe this method is suitable for these data. I think the PC analysis has identified instances where questions are highly similar. For example PC2 only contains items about spread, but I don’t think this is informative.

Referee Comments: Point-by-point response
NCOMMS-19-17863

Original Reviewer #2 Comments and Responses

1. The authors have included many items in their PCA that appear to be the same for example from Figure 1. “bread type: wholemeal/wholegrain vs. any other” and “bread type: wholemeal/wholegrain vs. white + brown” which correlate 0.9986 and in Figure 4 PC3 “butter and margarine spreads vs. oil-based spreads” and “spread type: butter and butter-like spreads vs. oil-based spreads” which show a correlation of 0.998. This seems to be true for a lot of the principal components in that they contain many highly correlated items. My question is to what extent is the PCs derived biased towards capturing variance from questions that are likely to be indistinguishable to most people i.e. spread type: butter + margarine vs. never vs spread type: butter vs. never?

Author response

We agree there is an extensive amount of phenotypic correlation between FI-QTs that were included in the PCA. While the addition of correlated variables will affect the total percentage of variance explained by each PC, it will not affect the resulting PCs and their respective combinations of foods. Therefore, we chose to take a more data-driven inclusive approach, and retain all quantitative traits in the PCA. This is yet another good reason to use heritability of each trait as a filter as opposed to a percentage of variance explained cutoff. We also hope including phenotypic correlation in Figure 1 and Supplementary Figure 4 allows the reader to view these correlated relationships within each PC. We have added the following text to the manuscript to explain this:

“While the inclusion of highly correlated FI-QTs in the PCA will affect the total percentage of variance explained by each PC, it will not affect the make-up of foods contributing to each PC. Therefore, we used heritability of all 85 FI-QTs and 85 PC-DPs as a filter for downstream genomic analysis.”

2. Furthermore, for many of these PCs (for example PC1, PC6, PC9, PC10, PC15, PC16, PC23, PC25, PC26, PC31, PC35, PC43) only a minority of items show a loading on the PCs greater than 5% indicating these are not contributing towards these components in a meaningful way. PC1 is discussed as “Overall, the FI-QTs that have high positive loadings for PC1 include wholemeal/wholegrain bread consumption, increased fruit and vegetable intake, increased oily fish intake, and increased water intake.” But that’s not really correct. It’s really a PC that captures whether white or brown bread is eaten. I think the authors could be clearer with what their PC’s are capturing.

Author response

It is the case that some of the PCs are largely driven by a small number of FI-QTs, and that including highly correlated FI-QTs will decrease their individual percent contribution to the PC. For instance, four highly correlated and overlapping bread-related FI-QTs each contribute 10-15% to PC1, together totally at most 60% contribution. The other 40% of PC1 is largely made up of the other FI-QTs that pass our significance threshold for inclusion in Figure 1. We use a simple cutoff for this in Figure 1 and S4 as described in the Methods section. (“FI-QTs with

percent contribution (squared coordinates) greater than expected under a uniform distribution [$1/85 \times 100 = 1.18\%$] were included in Figure 1 and Supplementary Figure 4”).

We believe we do in fact put appropriate emphasis on the contribution of type of bread on PC1 throughout the text as follows:

- “PC1 ... is primarily defined by the type of bread consumed (wholegrain/wholemeal vs. white bread). Overall, the FI-QTs that have high positive loadings for PC1 include wholemeal/wholegrain bread consumption, increased fruit and vegetable intake, increased oily fish intake, and increased water intake. The FI-QTs that have high negative loadings include white bread consumption, butter and oil spread consumption, increased processed meat intake, and consumption of milk with higher fat content (Figure 1).”
- “Finally, the aspects of a prudent dietary pattern reflected by PC1 (predominantly driven by wholemeal/wholegrain vs. white bread consumption) may not capture the causal protective features of a prudent dietary pattern.”

Furthermore, we include phenotypic correlation of the FI-QTs for each PC in Figure 1 and Supplementary Figure 4 to demonstrate the overlap among some of the FI-QTs.

Interestingly, when we repeat the MR analysis of educational attainment with the 19 FI-QTs that significantly contribute to PC1 and compare the results to educational attainment and PC1, educational attainment has the largest effect on PC1 (with non-overlapping CI with the 19 FI-QTs), indicating true effects on PC1 beyond just bread consumption.

Weighted Median Results with Educational Attainment Exposure

Outcome	Estimate	CI lower	CI upper	P-value
PC1	0.823	0.751	0.894	6.59E-113
bread type: wholemeal/wholegrain vs. white + brown	0.397	0.362	0.433	1.97E-106
bread type: wholemeal/wholegrain vs. any other	0.366	0.332	0.401	2.59E-95
bread type: white vs. any other	-0.357	-0.389	-0.325	6.18E-107
bread type: white vs. wholemeal/wholegrain + brown	-0.363	-0.396	-0.330	1.10E-101
pieces of fresh fruit per day	0.081	0.050	0.113	4.18E-07
spread type: all spreads vs. never	-0.050	-0.071	-0.028	7.65E-06
spread type: butter + margarine vs. never	-0.090	-0.122	-0.057	4.91E-08
spread type: butter vs. never	-0.077	-0.118	-0.036	0.000232102
overall processed meat intake	-0.017	-0.047	0.012	0.252382614
milk type: skimmed, semi-skimmed, full cream (QT)	0.049	0.021	0.078	0.000685274
overall oily fish intake	0.141	0.111	0.172	9.30E-20
tablespoons of raw vegetables per day	0.015	-0.015	0.045	0.342731568
pieces of dried fruit per day	0.253	0.223	0.283	4.61E-62
spread type: any oil based spread vs.	-0.066	-0.111	-0.021	0.004317514

never				
spread type: butter vs. any other	0.032	-0.001	0.066	0.061011324
spread type: other oil-based spread vs. never	-0.167	-0.225	-0.109	1.86E-08
spread type: olive oil spread vs. never	0.015	-0.051	0.081	0.655263859
glasses of water per day	0.072	0.041	0.102	5.18E-06
tablespoons of cooked vegetables per day	0.047	0.017	0.076	0.002009479

Additional Referee Comments

Responses in blue

The authors write “While the addition of correlated variables will affect the total percentage of variance explained by each PC” this is the issue. The PC analysis attempts to explain a large number of variables with a smaller number of components based on how the variables correlate. By including multiple measures of the same item (e.g. bread) the components derived by the authors will be largely composed of these repeated items. The first component only explains 8.63% of the variance and this is almost entirely driven by four questions about bread. However, this component is often called western Vs Prudent diet which is a little misleading looking at how much bread contributes to this factor. Another effect of including multiple items that are measuring the same thing is that the resulting component will be biased towards this subset of highly correlated items and lower the loadings on other items. For example, if a principal component analysis was conducted on five items measuring bread consumption and one of fruit consumption, the resulting component would have an axis closer to the cluster of bread items and further from the fruit. Repeat the analysis with one bread item and one fruit item and the axis will be closer to the fruit item. This shows that what the authors believe when they state “it will not affect the resulting PCs” is not actually correct.

We now do not refer to PC1 as a Western vs. prudent dietary pattern, and instead make a single comparison to these previously identified dietary patterns while pointing out the difference in bread as follows:

“PC1, which explains 8.63% (Supplementary Figure 2) of the total phenotypic variance in FI-QTs, contains foods similar to those that make up previously described Western and prudent dietary factors, though our PC1 is primarily defined by the type of bread consumed (wholegrain/wholemeal vs. white bread).”

We have also reviewed the text to make sure we are emphasizing the bread component to PC1 throughout the text, and in particular the Mendelian randomization section.

The reviewer raises an important limitation to including all FI-QTS in the PCA. The goal of PCA was to convert, in a data-driven way, a set of questions with complex and sometimes strong correlations to a set of independent combinations of these questions. Inclusion of different sets of correlated questions may change the structure of these PCs, but will not substantially inflate heritability (other than by diminishing noise introduced by repeated measures of correlated phenotypes) nor will it exclude questions that significantly contribute to the variance in the total FFQ dataset. We have added a sentence on this in the Results section and a paragraph (which we include below) on the limitations of our PC-dietary patterns in the Discussion section. Future

work re-evaluating PC-dietary patterns that arise from a PCA after removing highly correlated foods would be of interest.

“Of note, PC-DPs described throughout our manuscript were derived from a data-driven and unbiased PCA that included all FI-QTs, including those that were highly correlated and based upon the same FFQ question. Though this approach will shift each PC towards any of its overrepresented correlated measures, as is seen with bread type which makes up 56.7% of PC1, it will not substantially inflate heritability (other than by diminishing noise introduced by repeated measures of correlated phenotypes) nor will it exclude questions that significantly contribute to the variance in the total FFQ dataset. This also suggests that some PC-DPs will simply capture correlation between FI-QTs from the same FFQ question. For example, PC2 is made up of 11 significant FI-QTs based on a single FFQ question relating to the use of butter and oil-based spreads (Supplementary Figure 4). On the other hand, some PC-DPs capture the correlation between a variety of FI-QTs and FFQ questions fairly equally, such as PC7 whose lead contributors (3-7% each) include fish and meat intake, alcohol intake, tea and coffee consumption, cereal intake, fruit and vegetable intake, cheese intake, and never eating sugar (Supplementary Figure 4). Not surprisingly, SNP heritability, the proportion of phenotypic variance explained by common genetic variation, and number of GWAS loci is larger for PC7, a much more diverse dietary pattern, than for PC2 (Supplementary Figure 3B). Notably, future work with dietary patterns derived from non-overlapping FFQ questions could uncover new biology not discovered in our data-driven approach.”

I also disagree when the authors state that “Overall, the FI-QTs that have high positive loadings for PC1 include wholemeal/wholegrain bread consumption, increased fruit and vegetable intake, increased oily fish intake, and increased water intake. The FI-QTs that have high negative loadings include white bread consumption, butter and oil spread consumption, increased processed meat intake, and consumption of milk with higher fat content (Figure 1).” Looking at Figure 1. Bread items contribute between 10 and 15% but each of the non-bread items contributes less than 2.5%, which is in no way a high contribution. Again describing this as a Western vs. prudent diet is inaccurate as, it’s just bread.

We have made several changes to the text when referring to PC1:

- We have replaced “high” contributing foods with “significant” contributing foods
- We have included the % contribution in the main text when describing the food make-up of PC1
- We have removed the western and prudent labels
- We have emphasized the contribution of bread type to PC1 throughout the text

The authors seem very keen to describe their results in terms of identifying these components but I don’t believe this method is suitable for these data. I think the PC analysis has identified instances where questions are highly similar. For example PC2 only contains items about spread, but I don’t think this is informative.

We agree that because of the inclusion of highly correlated FI-QTs in the PCA a large portion of our PCs are driven by FI-QTs from the same or highly related FFQ questions. As the reviewer points out, PC2 is a combined phenotype solely based on spread use (both use of spreads and butter vs. oil FI-QTs). Not surprisingly, PC2 has less heritability than other lower order PCs, several of which include multiple different foods. In future analyses, we will likely not focus on

PC2 as a “pattern”, but rather a composite phenotype of spread use. The reviewer points out the importance of clarifying the interpretation of our PCs.

Though PC1 is largely driven by bread type, we still find it interesting and compelling (see above) that GWAS and MR of PC1 when compared to bread type FI-QTS are distinct, and that MR results of individual non-bread components of PC1 show similar and significant results with educational attainment as do the bread questions, suggesting that, while smaller, the other foods contributing to PC1 have a significant impact on these analyses. We have added the MR table (included above in previous response to comments) as a Supplementary Table in the manuscript, with this in-text reference:

“While PC1 is largely driven by bread type, MR of educational attainment on PC1’s 19 contributing FI-QTs demonstrates significant causal effects on both bread and non-bread FI-QTs, suggesting that while smaller, the other foods contributing to PC1 have a significant impact (Supplementary Table 3).”

We believe our PCA approach identifies some dietary patterns which truly are made up of multiple foods and may be of particular interest. PC7, for example, is our third most heritable PC and the contributions of its top foods are as follows: non-oily fish (6.4%), lamb (5.7%), pork (5.5%), tea (5.3%), beef (4.9%), alcohol with meals (4.1%), oily fish (3.8%), champagne (3.8%), never eat sugar (3.7%), cereal (3.7%)...